# Graph Neural Networks Provably Benefit from Structural Information: A Feature Learning Perspective

## Abstract

Graph neural networks (GNNs) have shown remarkable capabilities in learning from graph-structured data, outperforming traditional multilayer perceptrons (MLPs) in numerous graph applications. Despite these advantages, there has been limited theoretical exploration into why GNNs are so effective, particularly from the perspective of feature learning. This study aims to address this gap by examining the role of graph convolution in feature learning theory under a specific data generative model. We undertake a comparative analysis of the optimization and generalization between two-layer graph convolutional networks (GCNs) and their convolutional neural network (CNN) counterparts. Our findings reveal that graph convolution significantly enhances the regime of low test error over CNNs. This highlights a substantial discrepancy between GNNs and MLPs in terms of generalization capacity, a conclusion further supported by our empirical simulations on both synthetic and real-world datasets.

## 1 Introduction

Graph neural networks (GNNs) have recently demonstrated remarkable capability in learning graph representations, yielding superior results across various downstream tasks, such as node classifications (Kipf & Welling, 2016a; Veličković et al., 2017; Hamilton et al., 2017), graph classifications (Xu et al., 2018; Gilmer et al., 2017; Lee et al., 2019; Yuan & Ji, 2020) and link predictions (Kipf & Welling, 2016b; Zhang & Chen, 2018; Kumar et al., 2020), etc. However, the theoretical understanding of why GNNs can achieve such success is still in its infancy. Compared to multilayer perceptron (MLPs), GNNs enhance representation learning with an added message passing operation (Zhou et al., 2020). Take graph convoluational network (GCN) (Kipf & Welling, 2016a) as an example, it aggregates a node's attributes with those of its neighbors through a *graph convolution* operation. This operation, which leverages the structural information (adjacency matrix) of graph data, forms the core distinction between GNNs and MLPs. Empirical evidence from three node classification tasks, as shown in Figure 1, suggests GCNs outperform MLPs. Motivated by the superior performance of GNNs, we pose a critical question about graph convolution:

*What role does graph convolution play during gradient descent training, and what mechanism enables a GCN to exhibit better generalization after training?*

Several recent studies have embarked on a theoretical exploration of graph convolution's role in GNNs. For instance, Baranwal et al. (2021) considered a setting of linear classification of data generated from a contextual stochastic block model (Deshpande et al., 2018). Their findings indicate that graph convolution extends the regime where data is linearly separable by a factor of approximately $1/\sqrt{D}$ compared to MLPs, with $D$ denoting a node's expected degree. Baranwal et al. (2023) further investigated the impact of graph convolutions in multi-layer networks, showcasing improved non-linear separability. While insightful, these studies assume the Bayes optimal classifier of GNNs, thereby missing a comprehensive characterization of the GNNs' optimization process. This leaves a notable gap in understanding of the optimization and generalization capabilities of GNNs, a gap that existing theoretical explorations have yet to adequately address.

To respond to the growing demand for a comprehensive theoretical understanding of graph convolution, we delve into the feature learning analysis (Cao et al., 2022; Allen-Zhu & Li, 2022) for graph neural networks. In our study, we introduce a data generation model—termed SNM-SBM—that combines a signal-noise model (Cao et al., 2022; Allen-Zhu & Li, 2020) for feature creation and a stochastic block model (Abbe et al., 2015) for graph construction. Our analysis is centered on the convergence and generalization attributes of two-layer graph convolution networks (GCNs) when trained via gradient descent, compared with the established outcomes for two-layer convo-

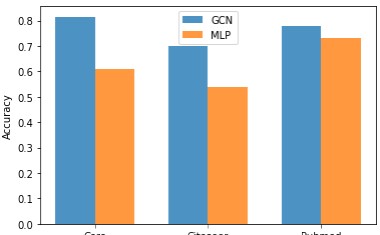

Figure 1: Performance comparison between GCN and MLP on node classification tasks.

lutional neural networks (CNNs) as presented by Cao et al. (2022). While both GCNs and CNNs demonstrate to achieve near-zero training error, our study effectively sheds light on the discrepancies in their generalization abilities. We emphasize the crucial contribution of graph convolution to the enhanced performance of GNNs. Our study's key contributions are as follows:

- We establish global convergence guarantees for graph neural networks training on data drawn from SNM-SBM model by characterizing the signal learning and noise memorization in feature learning. We demonstrate that, despite the nonconvex optimization landscape, GCNs can achieve zero training error after a polynomial number of iterations.

- We further establish population loss bounds of overfitted GNN models trained by gradient descent. We show that under certain conditions on the signal-to-noise ratio, GNNs trained by gradient descent can achieve near zero test error.

- We show a contrast in the generalization of GCNs and CNNs. We identify a regime where GCNs can attain nearly zero test error, whereas the test error of CNNs is greater than a constant. This conclusion is further supported by empirical verification on synthetic and real-world datasets.

## 2    RELATED WORK

**Role of Graph Convolution in GNNs.**    Enormous empirical studies of various GNNs models with graph convolution (Chen et al., 2017; Ma et al., 2021; Zhang et al., 2019; He et al., 2020; Wu et al., 2019; Wang et al., 2023) have been demonstrating that graph convolutions can enhance the performance of traditional classification methods, such as a multi-layer perceptron (MLP). Towards theoretically understanding the role of graph convolution, Xu et al. (2020) identify conditions under which MLPs and GNNs extrapolate, thereby highlighting the superiority of GNNs for extrapolation problems. Their theoretical analysis leveraged the concept of the over-parameterized networks and the neural tangent kernel (Jacot et al., 2018). Huang et al. (2021) use a similar approach to examine the role of graph convolution in deep GNNs within a node classification setting. They discover that excessive graph convolution layers can hamper the optimization and generalization of GNNs, corroborating the well-known over-smoothing issue in deep GNNs (Li et al., 2018). Another work by Hou et al. (2022) propose two smoothness metrics to measure the quantity and quality of information derived from graph data, along with a novel attention-based framework. Some rent works (Baranwal et al., 2021; 2023; Ma et al., 2021) have demonstrated that graph convolution broadens the regime in which a multi-layer network can classify nodes, compared to methods that do not utilize the graph structure, especially when the graph is dense and exhibits homophily. Yang et al. (2022) attribute the major performance gains of GNNs to their inherent generalization capability through graph neural tangent kernel (GNTK) and extrapolation analysis . As for neural network theory, these works either gleaned insights from GNTK (Du et al., 2019; Huang et al., 2021; Sabanayagam et al., 2022) or studied the role of graph convolution within a linear neural network setting. Unlike them, our work is beyond NTK and investigates a more realistic setting concerning the convergence and generalization of neural networks in terms of feature learning.

**Feature Learning in Neural Networks.**    This work builds upon a growing body of research on how neural networks learn features. Allen-Zhu & Li (2020) formulated a theory illustrating that when data possess a "multi-view" feature, ensembles of independently trained neural networks can demonstrably improve test accuracy. Further, Allen-Zhu & Li (2022) demonstrated that adversarial

training can purge certain small dense mixtures from the hidden weights during the training process of a neural network, thus refining the hidden weights. Ba et al. (2022) established that the initial gradient update contains a rank-1 'spike', which leads to an alignment between the first-layer weights and the linear component feature of the teacher model. Cao et al. (2022) investigated the benign overfitting phenomenon in training a two-layer convolutional neural network (CNN), illustrating that under certain conditions related to the signal-to-noise ratio, a two-layer CNN trained by gradient descent can achieve exceedingly low test loss through feature learning. Alongside related works (Yang & Hu, 2020; Zou et al., 2021; Wen & Li, 2021; Damian et al., 2022; Zou et al., 2023; Chen et al., 2023; Meng et al., 2023; Jelassi et al., 2022; Kou et al., 2023), all these studies have highlighted the existence of feature learning in neural networks during gradient descent training, forming a critical line of inquiry that this work continues to explore.

## 3 PROBLEM SETUP AND PRELIMINARY

### 3.1 NOTATIONS

We use lower bold-faced letters for vectors, upper bold-faced letters for matrices, and non-bold-faced letters for scalars. For a vector $\mathbf{v}$, its $\ell_2$-norm is denoted as $\|\mathbf{v}\|_2$. For a matrix $\mathbf{A}$, we use $\|\mathbf{A}\|_2$ to denote its spectral norm and $\|\mathbf{A}\|_F$ for its Frobenius norm. We employ standard asymptotic notations such as $O(\cdot), o(\cdot), \Omega(\cdot)$, and $\Theta(\cdot)$ to describe the limiting behavior. We use $\widetilde{O}(\cdot), \widetilde{\Omega}(\cdot)$, and $\widetilde{\Theta}(\cdot)$ to hide logarithmic factors in these notations respectively. Moreover, we denote $a_n = \text{poly}(b_n)$ if $a_n = O((b_n)^p)$ for some positive constant $p$ and $a_n = \text{polylog}(b_n)$ if $a_n = \text{poly}(\log(b_n))$. Lastly, sequences of integers are denoted as $[m] = \{1, 2, \ldots, m\}$.

### 3.2 DATA MODEL

In our approach, we utilize a signal-noise model for feature generation, combined with a stochastic block model for graph structure generation. Specifically, we define the feature matrix as $\mathbf{X} \in \mathbb{R}^{n \times 2d}$, with $n$ representing the number of samples and $2d$ being the feature dimensionality. Each feature associated with a data point is generated from a *signal-noise model* (SNM), conditional on the Rademacher random variable $y \in \{-1, 1\}$, and a latent vector $\boldsymbol{\mu} \in \mathbb{R}^d$:

$$\mathbf{x} = [\mathbf{x}^{(1)}, \mathbf{x}^{(2)}] = [y\boldsymbol{\mu}, \boldsymbol{\xi}], \tag{1}$$

where $\mathbf{x}^{(1)}, \mathbf{x}^{(2)} \in \mathbb{R}^d$, and $\boldsymbol{\xi} \sim \mathcal{N}(\mathbf{0}, \sigma_p^2 \cdot (\mathbf{I} - \|\boldsymbol{\mu}\|_2^{-2} \cdot \boldsymbol{\mu}\boldsymbol{\mu}^\top))$ is a Gaussian with $\sigma_p^2$ as the variance. The term $\mathbf{I} - \|\boldsymbol{\mu}\|_2^{-2} \cdot \boldsymbol{\mu}\boldsymbol{\mu}^\top$ is employed to guarantee that the noise vector is orthogonal to the signal vector $\boldsymbol{\mu}$. The signal-noise model we have adopted is inspired by the structure of an image composed of multiple patches, where we consider a two-patch model for simplicity. The first patch $\mathbf{x}^{(1)}$, represented by the signal vector, corresponds to the target in an image. The second patch $\mathbf{x}^{(2)}$, represented by the noise vector, corresponds to the background. It's worth mentioning that a series of recent works (Allen-Zhu & Li, 2020; Cao et al., 2022; Zou et al., 2021; Shen et al., 2022) have explored similar signal-noise models to illustrate the feature learning process of neural networks.

Moreover, we implement a stochastic block model with inter-class edge probability $p$ and intra-class edge probability $s$. Specifically, the entry of adjacency matrix $\mathbf{A} = (a_{ij})_{n \times n}$ is Bernoulli distributed, with $a_{ij} \sim \text{Ber}(p)$ when $y_i = y_j$, and $a_{ij} \sim \text{Ber}(s)$ when $y_i = -y_j$. The combination of a stochastic block model with the signal-noise model (1) is represented as $\text{SNM} - \text{SBM}(n, p, s, \boldsymbol{\mu}, \sigma_p, d)$. Note that when $p = s = 0$, $\text{SNM} - \text{SBM}$ reduces to a SNM, and its samples are used in MLP. In the SBM framework, the inter-class probability $p$ and intra-class probability $s$ are explicitly modeled, allowing us to analyze different graph structures based on the relationship between $p$ and $s$. When $p$ is significantly greater than $s$, the graph structure exhibits homophily. This means that the labels of neighboring nodes are likely to be similar to the label of the central node. Conversely, a heterophily graph structure is observed when $s$ is significantly greater than $p$. In this situation, nodes are more likely to connect with nodes of different labels.

### 3.3 NEURAL NETWORK MODEL

In this section, we present two distinct types of neural network models: a two-layer convolutional neural network (CNN), and a Graph Convolutional Neural Network (GCN) (Kipf & Welling, 2016a).

**CNN.** We introduce a two-layer CNN model, denoted as $f$, which utilizes a non-linear activation function, $\sigma(\cdot)$. Specifically, we employ a polynomial ReLU activation function defined as $\sigma(z) = \max\{0, z\}^q$, where $q > 2$ is a hyperparameter. Mathematically, given the input data $\mathbf{x}$, the CNN's output is represented as $f(\mathbf{W}, \mathbf{x}) = F_{+1}(\mathbf{W}_{+1}, \mathbf{x}) - F_{-1}(\mathbf{W}_{-1}, \mathbf{x})$, where $F_{+1}(\mathbf{W}_{+1}, \mathbf{x})$ and $F_{-1}(\mathbf{W}_{+1}, \mathbf{x})$ are defined as follows:

$$F_j(\mathbf{W}_j, \mathbf{x}) = \frac{1}{m} \sum_{r=1}^{m} \left[ \sigma(\mathbf{w}_{j,r}^\top \mathbf{x}^{(1)}) + \sigma(\mathbf{w}_{j,r}^\top \mathbf{x}^{(2)}) \right], \tag{2}$$

where $m$ is the width of hidden layer, the second layer parameters are fixed as either $+1$ or $-1$, and $\mathbf{w}_{j,r} \in \mathbb{R}^d$ refers to the weight of the first layer's $r$-th. The symbol $\mathbf{W}$ collectively represents the model's weights. Moreover, each weight in the first layer is initialized from a random draw of a Gaussian random variable, $\mathbf{w}_{j,r} \sim \mathcal{N}(\mathbf{0}, \sigma_0^2 \cdot \mathbf{I}_{d \times d})$ for all $r \in [m]$ and $j \in \{-1, 1\}$, with $\sigma_0$ regulating the initialization magnitude for the first layer's weight.

Upon receiving training data $\mathcal{S} \triangleq \{\mathbf{x}_i, y_i\}_{i=1}^n$ drawn from $\mathrm{SNM} - \mathrm{SBM}(n, p = 0, s = 0, \boldsymbol{\mu}, \sigma_p, d)$, we aim to learn the parameter $\mathbf{W}$ by minimizing the empirical cross-entropy loss function:

$$L_\mathcal{S}^{\mathrm{CNN}}(\mathbf{W}) = \frac{1}{n} \sum_{i=1}^{n} \ell(y_i \cdot f(\mathbf{W}, \mathbf{x}_i)), \tag{3}$$

where $\ell(y \cdot f(\mathbf{W}, \mathbf{x})) = \log(1 + \exp(-f(\mathbf{W}, \mathbf{x}) \cdot y))$. The update rule for the gradient descent used in the CNN is then given as:

$$\mathbf{w}_{j,r}^{(t+1)} = \mathbf{w}_{j,r}^{(t)} - \eta \cdot \nabla_{\mathbf{w}_{j,r}} L_S^{\mathrm{CNN}}(\mathbf{W}^{(t)})$$
$$= \mathbf{w}_{j,r}^{(t)} - \frac{\eta}{nm} \sum_{i=1}^{n} \ell_i'^{(t)} \cdot \sigma'(\langle \mathbf{w}_{j,r}^{(t)}, \boldsymbol{\xi}_i \rangle) \cdot j y_i \boldsymbol{\xi}_i - \frac{\eta}{nm} \sum_{i=1}^{n} \ell_i'^{(t)} \cdot \sigma'(\langle \mathbf{w}_{j,r}^{(t)}, y_i \boldsymbol{\mu} \rangle) \cdot j \boldsymbol{\mu}, \tag{4}$$

where we define the loss derivative as $\ell_i' \triangleq \ell'(y_i \cdot f_i) = -\frac{\exp(-y_i \cdot f_i)}{1 + \exp(-y_i \cdot f_i)}$. It's important to clarify that the model we use for the MLP part is a CNN. We categorize it as an MLP for comparison purposes.

**GCN.** Graph neural network (GNNs) fuse graph structure information and node features to learn representation of nodes. Consider a two-layer GCN $f$ with graph convolution operation on the first layer. The output of the GCN is given by $f(\mathbf{W}, \tilde{\mathbf{x}}) = F_{+1}(\mathbf{W}_{+1}, \tilde{\mathbf{x}}) - F_{-1}(\mathbf{W}_{-1}, \tilde{\mathbf{x}})$, where $F_{+1}(\mathbf{W}_{+1}, \tilde{\mathbf{x}})$ and $F_{-1}(\mathbf{W}_{+1}, \tilde{\mathbf{x}})$ are defined as follows:

$$F_j(\mathbf{W}_j, \tilde{\mathbf{x}}) = \frac{1}{m} \sum_{r=1}^{m} \left[ \sigma(\mathbf{w}_{j,r}^\top \tilde{\mathbf{x}}^{(1)}) + \sigma(\mathbf{w}_{j,r}^\top \tilde{\mathbf{x}}^{(2)}) \right]. \tag{5}$$

Here, $\tilde{\mathbf{X}} \triangleq [\tilde{\mathbf{x}}_1, \tilde{\mathbf{x}}_2, \cdots, \tilde{\mathbf{x}}_n]^\top = \tilde{\mathbf{D}}^{-1} \tilde{\mathbf{A}} \mathbf{X} \in \mathbb{R}^{n \times 2d}$ with $\tilde{\mathbf{A}} = \mathbf{A} + \mathbf{I}_n$ representing the adjacency matrix with self-loop, and $\tilde{\mathbf{D}}$ is a diagonal matrix that records the degree of each node, namely, $\tilde{D}_{ii} = \sum_j \tilde{A}_{ij}$. For simplicity we denote $D_i \triangleq \tilde{D}_{ii}$. Therefore, in contrast to the CNN model (2), the GCNs (5) incorporate the normalized adjacency matrix $\tilde{\mathbf{D}}^{-1} \tilde{\mathbf{A}}$, also termed as graph convolution, which serves as a pivotal component.

Note that the use of a polynomial ReLU activation function and fixing the second layer aligns with related studies (Allen-Zhu & Li, 2020; 2022; Cao et al., 2022; Zou et al., 2021; Kou et al., 2023) that investigate neural network feature learning.

With the training data $\mathcal{S} \triangleq \{\mathbf{x}_i, y_i\}_{i=1}^n$ and $\mathbf{A} \in \mathbb{R}^{n \times n}$ drawn from $\mathrm{SNM} - \mathrm{SBM}(n, p, s, \boldsymbol{\mu}, \sigma_p, d)$, we consider to learn the parameter $\mathbf{W}$ by optimizing the empirical cross-entropy loss function:

$$L_\mathcal{S}^{\mathrm{GCN}}(\mathbf{W}) = \frac{1}{n} \sum_{i=1}^{n} \ell(y_i \cdot f(\mathbf{W}, \tilde{\mathbf{x}}_i)). \tag{6}$$

The gradient descent update for the first layer weight $\mathbf{W}$ in GCN can be expressed as:

$$\mathbf{w}_{j,r}^{(t+1)} = \mathbf{w}_{j,r}^{(t)} - \eta \cdot \nabla_{\mathbf{w}_{j,r}} L_\mathcal{S}^{\mathrm{GCN}}(\mathbf{W}^{(t)})$$
$$= \mathbf{w}_{j,r}^{(t)} - \frac{\eta}{nm} \sum_{i=1}^{n} \ell_i'^{(t)} \sigma'(\langle \mathbf{w} b_{j,r}^{(t)}, \tilde{y}_i \boldsymbol{\mu} \rangle) \cdot j \tilde{y}_i \boldsymbol{\mu} - \frac{\eta}{nm} \sum_{i=1}^{n} \ell_i'^{(t)} \sigma'(\langle \mathbf{w}_{j,r}^{(t)}, \tilde{\boldsymbol{\xi}}_i \rangle) \cdot j y_i \tilde{\boldsymbol{\xi}}_i, \tag{7}$$

where we define "aggregated label" $\tilde{y}_i = D_i^{-1} \sum_{k \in \mathcal{N}(i)} y_k$ and "aggregated noise vector" $\tilde{\boldsymbol{\xi}}_i = D_i^{-1} \sum_{k \in \mathcal{N}(i)} \boldsymbol{\xi}_k$, with $\mathcal{N}(i)$ being a set that contains all the neighbor of node $i$.

In this study, our primary objective is to demonstrate the enhanced feature learning capabilities of GNNs in comparison to CNNs. This is achieved by examining the generalization ability of the GNN model through the lens of test error (population loss), which is defined based on unseen test data. Given $n$ training data points and the corresponding graph structure, we train a GNN model. We then generate a new test data point following the $\mathrm{SNM} - \mathrm{SBM}$ distribution. Its connection in the graph to the training data points are still following the stochastic block model, forming an adjacency matrix $\mathbf{A}' \in \mathbb{R}^{(n+1) \times (n+1)}$. We specifically study the population loss by taking the expectation over the randomness of the new test data, which is formulated as follows:

$$L_{\mathcal{D}}^{\mathrm{GCN}}(\mathbf{W}) = \mathbb{E}_{(\mathbf{x}, y, \mathbf{A}') \sim \mathrm{SNM} - \mathrm{SBM}} \ell(y \cdot f(\mathbf{W}, \mathbf{x})). \tag{8}$$

## 4 THEORETICAL RESULTS

In this section, we introduce our key theoretical findings that explain the optimization and generalization processes of feature learning in GCNs. Through the application of the gradient descent rule outlined in Equation (7), we observe that the gradient descent iterate $\mathbf{w}_{j,r}^{(t)}$ is a linear combination of its random initialization $\mathbf{w}_{j,r}^{(0)}$, the signal vector $\boldsymbol{\mu}$ and the noise vectors in the training data $\boldsymbol{\xi}_i$[1] for $i \in [n]$ (Cao et al., 2022). Consequently, for $r \in [m]$, the decomposition of weight can be expressed:

$$\mathbf{w}_{j,r}^{(t)} = \mathbf{w}_{j,r}^{(0)} + j \cdot \gamma_{j,r}^{(t)} \cdot \|\boldsymbol{\mu}\|_2^{-2} \cdot \boldsymbol{\mu} + \sum_{i=1}^{n} \overline{\rho}_{j,r,i}^{(t)} \cdot \|\boldsymbol{\xi}_i\|_2^{-2} \cdot \boldsymbol{\xi}_i + \sum_{i=1}^{n} \underline{\rho}_{j,r,i}^{(t)} \cdot \|\boldsymbol{\xi}_i\|_2^{-2} \cdot \boldsymbol{\xi}_i. \tag{9}$$

where $\gamma_{j,r}^{(t)}$ and $\rho_{j,r,i}^{(t)} = \{\overline{\rho}_{j,r,i}^{(t)}, \underline{\rho}_{j,r,i}^{(t)}\}$ serve as coefficients. To facilitate a fine-grained analysis for the evolution of coefficients, we introduce the notations $\overline{\rho}_{j,r,i}^{(t)} \triangleq \rho_{j,r,i}^{(t)} \mathbb{1}(\rho_{j,r,i}^{(t)} \geq 0)$, $\underline{\rho}_{j,r,i}^{(t)} \triangleq \rho_{j,r,i}^{(t)} \mathbb{1}(\rho_{j,r,i}^{(t)} \leq 0)$. We refer to Equation (9) as the signal-noise decomposition of $\mathbf{w}_{j,r}^{(t)}$. The normalization factors $\|\boldsymbol{\mu}\|_2^{-2}$ and $\|\boldsymbol{\xi}_i\|_2^{-2}$ are introduced to ensure that $\gamma_{j,r}^{(t)} \approx \langle \mathbf{w}_{j,r}^{(t)}, \boldsymbol{\mu} \rangle$, and $\rho_{j,r,i}^{(t)} \approx \langle \mathbf{w}_{j,r}^{(t)}, \boldsymbol{\xi}_i \rangle$. We employ $\gamma_{j,r}^{(t)}$ to characterize the process of signal learning and $\rho_{j,r,i}^{(t)}$ to characterize the noise memorization. If certain $\gamma_{j,r}^{(t)}$ values are sufficiently large while all $|\rho_{j,r,i}^{(t)}|$ are relatively small, this indicates that the neural network is primarily learning the label through signle learning. Conversely, if some $|\rho_{j,r,i}^{(t)}|$ values are relatively large while all $\gamma_{j,r}^{(t)}$ are small, the neural network will focus on noise memorization. Our analysis is based on the following assumptions:

**Assumption 4.1.** *Suppose that*

1. *The dimension $d$ is sufficiently large: $d = \tilde{\Omega}(m^{2 \vee [4/(q-2)]} n^{4 \vee [(2q-2)/(q-2)]})$.*

2. *The size of training sample $n$ and width of GCNs $m$ adhere to $n, m = \Omega(\mathrm{polylog}(d))$.*

3. *The learning rate $\eta$ satisfies $\eta \leq \tilde{O}(\min\{\|\boldsymbol{\mu}\|_2^{-2}, \sigma_p^{-2} d^{-1}\})$.*

4. *The edge probability $p, s = \Omega(\sqrt{\log(n)/n})$ and $\Xi \triangleq \frac{p-s}{p+s}$ is a positive constant.*

5. *The standard deviation of Gaussian initialization $\sigma_0$ is chosen such that $\sigma_0 \leq \tilde{O}(m^{-2/(q-2)} n^{-[1/(q-2)] \vee 1} \cdot \min\{(\sigma_p \sqrt{d/(n(p+s))})^{-1}, \Xi^{-1} \|\boldsymbol{\mu}\|_2^{-1}\})$.*

**Remark 4.2.** *(1) The requirement for a high dimension in our assumptions is specifically aimed at ensuring that the learning occurs in a sufficiently over-parameterized setting when the second layer remains fixed. (2) This condition ensures certain statistical properties of the training data and weight initialization hold with a probability of at least $1 - d^{-1}$. (3) The condition on $\eta$ is to ensure that gradient descent can effectively minimize the training loss. (4) The assumption regarding edge probability guarantees a sufficient level of concentration in the degree and an adequate display*

---

[1]By referring to Equation (7), we assert that the gradient descent update moves in the direction of $\tilde{\boldsymbol{\xi}}_i$ for each $i \in [n]$. Then we can apply the definition of $\tilde{\boldsymbol{\xi}}_i = D_i^{-1} \sum_{k \in \mathcal{N}(i)} \boldsymbol{\xi}_k$.

*of homophily of graph data. (5) Lastly, the conditions imposed on initialization strength $\sigma_0$ are intended to guarantee that the training loss can effectively converge to a sufficiently small value and to discern the differential learning speed between signal and noise.*

Finally, we introduce a critical quantity called signal-to-noise ratio (SNR), which can measure the relative learning speed between signal and noise, as is calculated through $\text{SNR} = \|\boldsymbol{\mu}\|_2/(\sigma_p\sqrt{d})$. To prepare for our main result, we provide an effective SNR for GNNs, defend as $\text{SNR}_G = \|\boldsymbol{\mu}\|_2/(\sigma_p\sqrt{d}) \cdot (n(p+s))^{(q-2)/(2q)}$. Given the above assumptions and definitions of SNR, we present our main result for GNN as follows:

**Theorem 4.3.** *Let $T = \tilde{\Theta}(\eta^{-1}m\sigma_0^{-(q-2)}\Xi^{-q}\|\boldsymbol{\mu}\|_2^{-q} + \eta^{-1}\epsilon^{-1}m^3\|\boldsymbol{\mu}\|_2^{-2})$. Under Assumption 4.1, if $n \cdot \text{SNR}_G^q = \tilde{\Omega}(1)$, then with probability at least $1 - d^{-1}$, there exists a $0 \le t \le T$ such that:*

- *The GCN learns the signal: $\max_r \gamma_{j,r}^{(t)} = \tilde{\Omega}(1)$ for $j \in \{\pm 1\}$.*

- *The GCN does not memorize the noises in the training data: $\max_{j,r,i} |\rho_{j,r,i}^{(T)}| = \tilde{O}(\sigma_0\sigma_p\sqrt{d/n(p+s)})$.*

- *The training loss converges to $\epsilon$, i.e., $L_S^{\text{GCN}}(\mathbf{W}^{(t)}) \le \epsilon$.*

- *The trained GCN achieves a small test loss: $L_{\mathcal{D}}^{\text{GCN}}(\mathbf{W}^{(t)}) \le c_1\epsilon + \exp(-c_2n^2)$.*

*where $c_1$ and $c_2$ are positive constants.*

Theorem 4.3 reveals that, provided $n \cdot \text{SNR}_G^q = \tilde{\Omega}(1)$, the GCN can learn the signal by achieving $\max_r \gamma_{j,r}^{(t)} = \Omega(1)$, and on the other hand, the noise memorization during gradient descent training is suppressed by $\max_{j,r,i} |\rho_{j,r,i}^{(T)}| = \tilde{O}(\sigma_0\sigma_p\sqrt{d/n(p+s)})$, given that $\sigma_0\sigma_p\sqrt{d/n(p+s)} \ll 1$ according to assumption 4.1. Because the signal learned by the network is large enough and much stronger than the noise memory, it can generalize well to test sample. Consequently, the learned neural network can achieve both small training and test losses. It's worth noting that when the graph's degree is reduced to 1, the effective SNR for GNNs converges to the vanilla SNR, namely $\text{SNR}_G = \text{SNR}$. This reduces to the CNN, whose feature learning is established by Cao et al. (2022).

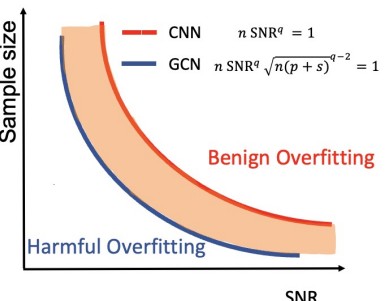

Figure 2: Illustration of performance comparison between GNN and CNN. The orange band highlights where GNN can outperform CNN.

Our result show that whether a neural network learns the signal or noise depends on the SNR, and the number of samples $n$. According to Cao et al. (2022), CNNs can focus on the signal learning and generalize well on the unseen data when $n \cdot \text{SNR}^q = \tilde{\Omega}(1)$. On the other hand, when $n \cdot \text{SNR}^q = \tilde{O}(1)$, CNNs mainly memorize the noise from data, thus achieve a large test error. To highlight the differences in generalization between GNNs and CNNs, we show that, if $n \cdot \text{SNR}_G^q = \tilde{\Omega}(1)$ and $n \cdot \text{SNR}^q = \tilde{O}(1)$, then the trained **GNNs achieve small test error**, given by $L_{\mathcal{D}}^{\text{GCN}}(\mathbf{W}^{(t)}) = o(1)$. In contrast, the trained **CNNs achieve large test error**, $L_{\mathcal{D}}^{\text{CNN}}(\mathbf{W}^{(t)}) \ge C$. The first condition $n \cdot \text{SNR}_G^q = \tilde{\Omega}(1)$ is by Theorem 4.3, while second the condition $n \cdot \text{SNR}_G^q = \tilde{\Omega}(1)$ is based on the findings of Cao et al. (2022) for CNN. As a conclusion, we clearly provide a condition that GNNs can generalize better than CNNs. This observation is further visualized in Figure 2. Through the precise characterization of feature learning from optimization to generalization for GNN, we have successfully demonstrated that the graph neural network can gain superiority with the help of graph convolution.

## 5 PROOF SKETCHES

In this section, we present proof sketches inspired by the study of feature learning in CNNs (Cao et al., 2022). This foundation allows us to extend and adapt these concepts to a novel context for

GNNs. We discuss the primary challenges encountered during the study of GNN, and illustrate the key techniques we employed in our proofs to overcome these challenges. These main techniques are elaborated in the following sections, and detailed proofs can be found in the appendix.

## 5.1 ITERATIVE OF COEFFICIENTS UNDER GRAPH CONVOLUTION

To analyze the feature learning process of graph neural networks during gradient descent training, we introduce an iterative methodology, based on the signal-noise decomposition in decomposition (9) and gradient descent update (7). The following lemma offers us a means to monitor the iteration of the signal learning and noise memorization under graph convolution:

**Lemma 5.1.** *The coefficients* $\gamma_{j,r}^{(t)}, \overline{\rho}_{j,r,i}^{(t)}, \underline{\rho}_{j,r,i}^{(t)}$ *in decomposition (9) adhere to the following equations:*

$$\gamma_{j,r}^{(0)}, \overline{\rho}_{j,r,i}^{(0)}, \underline{\rho}_{j,r,i}^{(0)} = 0, \tag{10}$$

$$\gamma_{j,r}^{(t+1)} = \gamma_{j,r}^{(t)} - \frac{\eta}{nm} \cdot \sum_{i=1}^{n} \ell_i'^{(t)} \sigma'(\langle \mathbf{w}_{j,r}^{(t)}, \tilde{y}_i \boldsymbol{\mu}_i \rangle) y_i \tilde{y}_i \|\boldsymbol{\mu}\|_2^2, \tag{11}$$

$$\overline{\rho}_{j,r,i}^{(t+1)} = \overline{\rho}_{j,r,i}^{(t)} - \frac{\eta}{nm} \cdot \sum_{k \in \mathcal{N}(i)} D_k^{-1} \cdot \ell_k'^{(t)} \cdot \sigma'(\langle \mathbf{w}_{j,r}^{(t)}, \tilde{\boldsymbol{\xi}}_k \rangle) \cdot \|\boldsymbol{\xi}_i\|_2^2 \cdot \mathbb{1}(y_k = j), \tag{12}$$

$$\underline{\rho}_{j,r,i}^{(t+1)} = \underline{\rho}_{j,r,i}^{(t)} + \frac{\eta}{nm} \cdot \sum_{k \in \mathcal{N}(i)} D_k^{-1} \cdot \ell_k'^{(t)} \cdot \sigma'(\langle \mathbf{w}_{j,r}^{(t)}, \tilde{\boldsymbol{\xi}}_k \rangle) \cdot \|\boldsymbol{\xi}_i\|_2^2 \cdot \mathbb{1}(y_k = -j). \tag{13}$$

Lemma 5.1 simplifies the analysis of the feature learning in GCNs by reducing it to the examination of the discrete dynamical system expressed by Equations (11 - 13). Our proof strategy emphasizes an in-depth evaluation of the coefficient values $\gamma_{j,r}^{(t)}, \overline{\rho}_{j,r,i}^{(t)}, \underline{\rho}_{j,r,i}^{(t)}$ throughout the training. Note that graph convolution aggregates information from neighboring nodes to the central node, which often leads to the loss of statistical stability for the aggregated noise vectors and labels. To overcome this challenge, we utilize a dense graph input, achieved by setting the edge probability as stated in 4.1.

## 5.2 A TWO-PHASE DYNAMICS ANALYSIS

We then provide a two-stage dynamics analysis based on the behavior of loss derivative to track the trajectory of coefficients for signal learning and noise memorization:

**Stage 1.** Intuitively, the initial neural network weights are small enough so that the neural network at initialization has constant level cross-entropy loss derivatives on all the training data: $\ell_i'^{(0)} = \ell'[y_i \cdot f(\mathbf{W}^{(0)}, \tilde{\mathbf{x}}_i)] = \Theta(1)$ for all $i \in [n]$. This is guaranteed under Condition 4.1 on $\sigma_0$. Motivated by this, the dynamics of the coefficients in Equations (11 - 13) can be greatly simplified by replacing the $\ell_i'^{(t)}$ factors by their constant upper and lower bounds. The following lemma summarizes our main conclusion at stage 1 for signal learning:

**Lemma 5.2.** *Under the same conditions as Theorem 4.3, there exists $T_1 = \tilde{O}(\eta^{-1} m \sigma_0^{2-q} \Xi^{-q} \|\boldsymbol{\mu}\|_2^{-q})$ such that $\max_r \gamma_{j,r}^{(T_1)} = \Omega(1)$ for $j \in \{\pm 1\}$, and $|\rho_{j,r,i}^{(t)}| = O\left(\sigma_0 \sigma_p \sqrt{d}/\sqrt{n(p+s)}\right)$ for all $j \in \{\pm 1\}$, $r \in [m]$, $i \in [n]$ and $0 \leq t \leq T_1$.*

The proof can be found in Appendix C.1. Lemmas 5.2 leverages the period of training when the derivatives of the loss function are of a constant order. It's important to note that graph convolution plays a significant role in diverging the learning speed between signal learning and noise memorization in this first stage. Note that graph convolution can potentially cause unstable iterative dynamics of coefficients during the feature learning process. To mitigate this issue, we introduce "homophily" by setting $p > s$, which helps in stabilizing the coefficient iterations.

Originally, the learning speeds are roughly determined by $\|\boldsymbol{\mu}\|_2$ and $\|\boldsymbol{\xi}\|_2$ respectively without graph convolution (Cao et al., 2022). Instead, with graph convolution, the learning speeds are approximately determined by $|\tilde{y}| \|\boldsymbol{\mu}\|_2$ and $\|\tilde{\boldsymbol{\xi}}\|_2$ respectively. Here, $|\tilde{y}| \|\boldsymbol{\mu}\|_2$ is close to $\|\boldsymbol{\mu}\|_2$, but $\|\tilde{\boldsymbol{\xi}}\|_2$ is much smaller than $\|\boldsymbol{\xi}\|_2$ (see Figure 6 for an illustration). This means that graph convolution can slow down noise memorization, thus enabling GNNs to focus more on signal learning.

**Stage 2.** Building on the results from the first stage, we then move to the second stage of the training process. In this stage, the loss derivatives are no longer constant, and we demonstrate that the training error can be minimized to an arbitrarily small value. Importantly, the scale differences established during the first stage of learning continue to be maintained throughout the second stage:

**Lemma 5.3.** *Under the same conditions as Theorem 4.3, for any $t \in [T_1, T]$, it holds that $\max_r \gamma_{j,r}^{(T_1)} \geq 2, \forall j \in \{\pm 1\}$ and $|\rho_{j,r,i}^{(t)}| \leq \sigma_0 \sigma_p \sqrt{d/(n(p+s))}$ for all $j \in \{\pm 1\}$, $r \in [m]$ and $i \in [n]$. Moreover, we have $L_{\mathcal{S}}^{\mathrm{GCN}}(\mathbf{W}^{(t)}) \leq \epsilon$.*

Lemma 5.3 presents two primary outcomes. Firstly, throughout this training phase, it ensures that the coefficients of noise vectors, denoted as $\rho_{j,r,i}^{(t)}$, retain a significantly small value while coefficients of feature vector, denoted as $\gamma_{j,r}^{(t)}$ can achieve large value. Furthermore, it offers a convergence for GNN, showing the training loss will tend to receive an arbitrarily small value.

### 5.3 TEST ERROR ANALYSIS

Finally, it is a challenge for the generalization analysis of graph neural networks. To address this issue, we introduce an expectation over the distribution for a single data point. We consider a new data point $(\mathbf{x}, y)$ drawn from the distribution SNM-SBM. The lemma below further gives an upper bound on the test loss of GNNs post-training:

**Lemma 5.4.** *Let $T$ be defined in Theorem 4.3. Under the same conditions as Theorem 4.3, for any $t \leq T$ with $L_{\mathcal{S}}^{\mathrm{GCN}}(\mathbf{W}^{(t)}) \leq 1$, it holds that $L_{\mathcal{D}}^{\mathrm{GCN}}(\mathbf{W}^{(t)}) \leq c_1 \cdot L_{\mathcal{S}}^{\mathrm{GCN}}(\mathbf{W}^{(t)}) + \exp(-c_2 n^2)$.*

The proof is presented in the appendix. Lemma 5.4 demonstrates that GNNs achieve a small test error (*benign overfitting*) and completes the last step of feature learning theory.

## 6 EXPERIMENTS

In this section, we validate our theoretical findings through numerical simulations using synthetic data, specifically generated according to the SNM-SBM model. We set the signal vector, $\boldsymbol{\mu}$, to drawn from a standard normal distribution $\mathcal{N}(\mathbf{0}, \mathbf{I})$. The noise vector, $\boldsymbol{\xi}$, is drawn from a Gaussian distribution $\mathcal{N}(\mathbf{0}, \sigma_p^2 \mathbf{I})$. We train a two-layer CNN defined as per equation (2) and a two-layer GNN as per equation (5) with polynomial ReLU $q = 3$. We used the gradient descent method with a learning rate of $\eta = 0.03$. The primary task we focused on was node classification, where the goal was to predict the class labels of nodes in a graph.

**Feature learning dynamics.** Firstly, we display the training loss, test loss, training accuracy, and test accuracy for both the CNN and GNN in Figure 3. In this case, we further set the training data size to $n = 250$, input dimension to $d = 500$, noise strength to $\sigma_p = 20$, and edge probability to $p = 0.5$, $s = 0.08$. We observe that both the GNN and CNN can achieve zero training error. However, while the GNN obtains nearly zero test error, the CNN fails to generalize effectively to the test set. This simulation result serves to validate our theoretical results in Theorem 4.3.

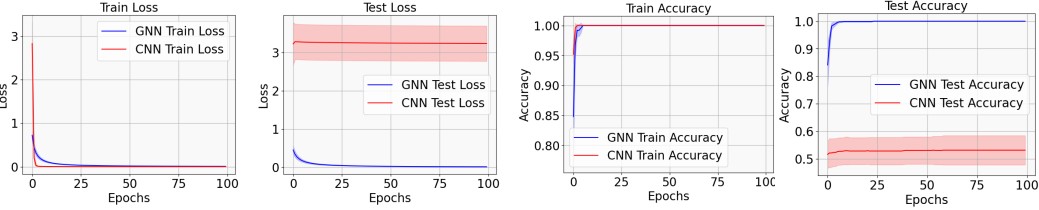

Figure 3: Training loss, testing loss, training accuracy, and testing accuracy for both CNN and GNN over a span of 100 training epochs. Five experimental runs are conducted, with shaded areas highlighting error bars for variability.

**Verification via real-world data.** We conducted an experiment using real-world data, specifically by replacing the synthetic feature with MNIST input features. We select numbers 1 and 2 from the ten digital numbers, and applied both CNN and GNN models as described in our paper. Detailed

results and visualizations can be found in the Figure 4. The results were consistent with our theoretical conclusions, reinforcing the insights derived from our analysis. We believe that this experiment adds a valuable dimension to our work, bridging the gap between theory and practice.

**Phase diagram.** We then explore a range of Signal-to-Noise Ratios (SNRs) from 0.045 to 0.98, and a variety of sample sizes, $n$, ranging from 200 to 7200. Based on our results, we train the neural network for 200 steps for each combination of SNR and sample size $n$. After training, we calculate the test accuracy for each run. The results are presented as a heatmap in Figure 5. Compared to CNNs, GCNs demonstrate a perfect accuracy score of 1 across a more extensive range in the SNR and $n$ plane, indicating that GNNs have a broader *benign overfitting* regime with high test accuracy. This further validates our theoretical findings.

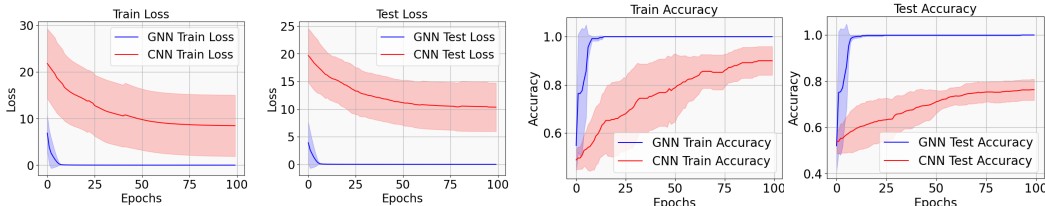

Figure 4: The verification of our theoretical result with a real-world data. The input feature is form MNIST dataset, where we select number 1 and 2 as two classes. The graph structure is sampled form stochastic block model, with edge probability $p = 0.75$, $s = 0.05$. We show the training loss, testing loss, training accuracy, and testing accuracy for both CNN and GNN over a span of 100 training epochs. The results confirm the benefit of GNN over CNN on the real world dataset. Five experimental runs are conducted, with shaded areas highlighting error bars for variability.

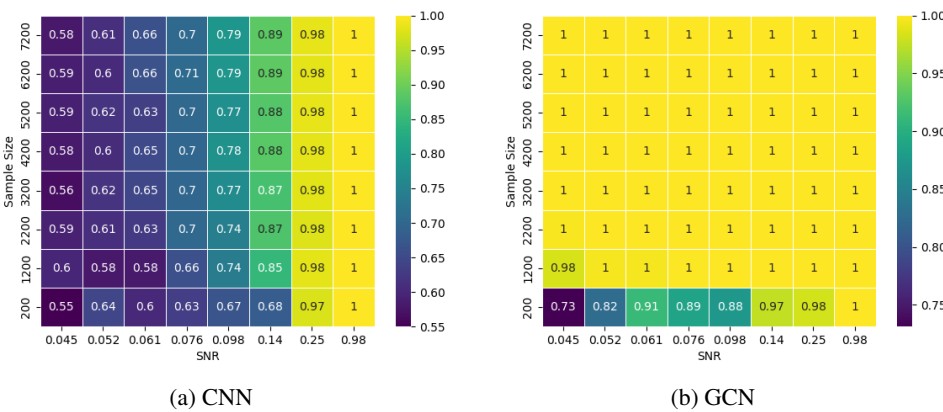

(a) CNN                                        (b) GCN

Figure 5: Test accuracy heatmap for CNNs and GCNs after training.

# 7 CONCLUSION AND LIMITATIONS

This paper utilizes a signal-noise decomposition to study the signal learning and noise memorization process in training a two-layer GCN. We provide specific conditions under which a GNN will primarily concentrate on signal learning, thereby achieving low training and testing errors. Our results theoretically demonstrate that GCNs, by leveraging structural information, outperform CNNs in terms of generalization ability across a broader benign regime. As a pioneering work that studies feature learning of GNNs, our theoretical framework is constrained to examining the role of graph convolution within a specific two-layer GCN and a certain data generalization model. In fact, the feature learning of a neural network can be influenced by a myriad of other factors, such as the depth of GNN, activation function, optimization algorithm, and data model (Kou et al., 2023; Zou et al., 2021; 2023). Future work can extend our framework to consider the influence of a wider array of factors on feature learning within GCNs.

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

## A  PRELIMINARY LEMMAS

In this section, we present preliminary lemmas which form the foundation for the proofs to be detailed in the subsequent sections. The proof will be developed after the lemmas presented.

### A.1  PRELIMINARY LEMMAS WITHOUT GRAPH CONVOLUTION

In this section, we introduce necessary lemmas that will be used in the analysis without graph convolution, following the study of feature learning in CNN (Cao et al., 2022). In particular, Lemma A.1 states that noise vectors are "almost orthogonal" to each other and Lemma A.2 indicates that random initialization results in a controllable inner product between the weights at initialization and the data vectors.

**Lemma A.1.** *(Cao et al., 2022) Suppose that $\delta > 0$ and $d = \Omega(\log(4n/\delta))$. Then with probability at least $1 - \delta$,*

$$\sigma_p^2 d/2 \leq \|\boldsymbol{\xi}_i\|_2^2 \leq 3\sigma_p^2 d/2,$$
$$|\langle \boldsymbol{\xi}_i, \boldsymbol{\xi}_{i'} \rangle| \leq 2\sigma_p^2 \cdot \sqrt{d \log(4n^2/\delta)},$$

*for all $i, i' \in [n]$.*

**Lemma A.2.** *(Cao et al., 2022) Suppose that $d = \Omega(\log(nm/\delta))$, $m = \Omega(\log(1/\delta))$. Then with probability at least $1 - \delta$,*

$$|\langle \mathbf{w}_{j,r}^{(0)}, \boldsymbol{\mu} \rangle| \leq \sqrt{2 \log(8m/\delta)} \cdot \sigma_0 \|\boldsymbol{\mu}\|_2,$$
$$|\langle \mathbf{w}_{j,r}^{(0)}, \boldsymbol{\xi}_i \rangle| \leq 2\sqrt{\log(8mn/\delta)} \cdot \sigma_0 \sigma_p \sqrt{d},$$

*for all $r \in [m]$, $j \in \{\pm 1\}$ and $i \in [n]$. Moreover,*

$$\sigma_0 \|\boldsymbol{\mu}\|_2/2 \leq \max_{r \in [m]} j \cdot \langle \mathbf{w}_{j,r}^{(0)}, \boldsymbol{\mu} \rangle \leq \sqrt{2 \log(8m/\delta)} \cdot \sigma_0 \|\boldsymbol{\mu}\|_2,$$
$$\sigma_0 \sigma_p \sqrt{d}/4 \leq \max_{r \in [m]} j \cdot \langle \mathbf{w}_{j,r}^{(0)}, \boldsymbol{\xi}_i \rangle \leq 2\sqrt{\log(8mn/\delta)} \cdot \sigma_0 \sigma_p \sqrt{d},$$

*for all $j \in \{\pm 1\}$ and $i \in [n]$.*

### A.2  PRELIMINARY LEMMAS ON GRAPH PROPERTIES

We now introduce important lemmas that are critical to our analysis. The key idea to ensure a relatively dense graph. In a sparser graph, the concentration properties of graph degree (Lemma A.3), the graph convoluted label (A.4), the graph convoluted noise vector (Lemma A.7 and Lemma A.5) are no longer guaranteed. This lack of concentration affects the behavior of coefficients during gradient descent training, leading to deviations from our current main results.

**Lemma A.3** (Degree concentration). *Let $p, s = \Omega\left(\sqrt{\frac{\log(n/\delta)}{n}}\right)$ and $\delta > 0$, then with probability at least $1 - \delta$, we have*

$$n(p + s)/4 \le D_i \le 3n(p + s)/4.$$

*Proof.* It is known that the degrees are sums of Bernoulli random variables.

$$D_i = 1 + \sum_{j \neq i}^{n} a_{ij},$$

where $a_{ij} = [\mathbf{A}]_{ij}$. Hence, by the Hoeffding's inequality, with probability at least $1 - \delta/n$

$$|D_i - \mathbb{E}[D_i]| < \sqrt{\log(n/\delta)(n-1)}.$$

Note that $a_{ii} = 1$ is a fixed value, which means that it is not a random variable, thus the denominator in the exponential part is $n - 1$ instead of $n$. Now we calculate the expectation of degree:

$$\mathbb{E}[D_{ii}] = 1 + \frac{n}{2}s + (\frac{n}{2} - 1)p = n(p + s)/2 + 1 - p,$$

then we have

$$|D_i - n(p + s)/2 + 1 - p| \le \sqrt{n \log(n/\delta)}.$$

Because that $p, s = \Omega\left(\sqrt{\frac{\log(n/\delta)}{n}}\right)$, we further have,

$$n(p + s)/4 \le D_i \le 3n(p + s)/4.$$

Applying a union bound over $i \in [n]$ conclude the proof. □

**Lemma A.4.** *Suppose that $\delta > 0$ and $n \ge 8\frac{p+s}{(p-s)^2}\log(4/\delta)$. Then with probability at least $1 - \delta$,*

$$\frac{1}{2}\frac{p - s}{p + s}|y_i| \le |\tilde{y}_i| \le \frac{3}{2}\frac{p - s}{p + s}|y_i|.$$

*Proof of Lemma A.4.* By Hoeffding's inequality, with probability at least $1 - \delta/2$, we have

$$\left| \frac{1}{D_i} \sum_{k \in \mathcal{N}(i)} y_k - \frac{p - s}{p + s}y_i \right| \le \sqrt{\frac{\log(4/\delta)}{2n(p + s)}}.$$

Therefore, as long as $n \ge 8\frac{p+s}{(p-s)^2}\log(4/\delta)$, we have:

$$\frac{1}{2}\frac{p - s}{p + s}|y_i| \le |\tilde{y}_i| \le \frac{3}{2}\frac{p - s}{p + s}|y_i|.$$

This proves the result for the stability of sign of graph convoluted label. □

**Lemma A.5.** *Suppose that $\delta > 0$ and $d = \Omega(n^2(p + s)^2 \log(4n^2/\delta))$. Then with probability at least $1 - \delta$,*

$$\sigma_p^2 d/(4n(p + s)) \le \|\tilde{\boldsymbol{\xi}}_i\|_2^2 \le 3\sigma_p^2 d/(4n(p + s)),$$

*for all $i \in [n]$.*

*Proof of Lemma A.5.* It is known that:

$$\|\tilde{\boldsymbol{\xi}}_i\|_2^2 = \frac{1}{D_i^2} \sum_{j=1}^{d} \left( \sum_{k=1}^{D_i} \xi_{jk} \right)^2 = \frac{1}{D_i^2} \sum_{j=1}^{d} \sum_{k=1}^{D_i} \xi_{jk}^2 + \frac{1}{D_i^2} \sum_{j=1}^{d} \sum_{k \neq k'}^{D_i} \xi_{jk'}\xi_{jk}.$$

By Bernstein's inequality, with probability at least $1 - \delta/(2n)$ we have

$$\left| \sum_{j=1}^{d} \sum_{k=1}^{D_i} \xi_{jk}^2 - \sigma_p^2 d D_i \right| = O(\sigma_p^2 \cdot \sqrt{dD_i \log(4n/\delta)}).$$

Therefore, as long as $d = \Omega(\log(4n/\delta)/(n(p+s)))$, we have

$$3\sigma_p^2 dD_i/4 \leq \sum_{j=1}^{d} \sum_{k=1}^{D_i} \xi_{jk}^2 \leq 5\sigma_p^2 dD_i/4.$$

By Lemma A.3, we have,

$$2\sigma_p^2 d/(4n(p+s)) \leq \frac{1}{D_i^2} \sum_{j=1}^{d} \sum_{k=1}^{D_i} \xi_{jk}^2 \leq 6\sigma_p^2 d/(4n(p+s)).$$

Moreover, clearly $\langle \boldsymbol{\xi}_k, \boldsymbol{\xi}_{k'} \rangle$ has mean zero. For any $k, k'$ with $k \neq k'$, by Bernstein's inequality, with probability at least $1 - \delta/(2n^2)$ we have

$$|\langle \boldsymbol{\xi}_k, \boldsymbol{\xi}_{k'} \rangle| \leq 2\sigma_p^2 \cdot \sqrt{d\log(4n^2/\delta)}.$$

Applying a union bound we have that with probability at least $1 - \delta$,

$$|\langle \boldsymbol{\xi}_k, \boldsymbol{\xi}_{k'} \rangle| \leq 2\sigma_p^2 \cdot \sqrt{d\log(4n^2/\delta)}.$$

Therefore, as long as $d = \Omega(n^2(p+s)^2 \log(4n^2/\delta))$, we have

$$\sigma_p^2 d/(4n(p+s)) \leq \|\tilde{\boldsymbol{\xi}}_i\|_2^2 \leq 3\sigma_p^2 d/(4n(p+s)).$$

**Remark A.6.** *We compare the noise vector both before and after applying graph convolution. By examining Lemma A.1 and Lemma A.5, we discover that the expectation of the $\ell_2$ norm of noise vector is reduced by a factor of $\sqrt{n(p+s)/2}$. This factor represents the square root of the expected degree of the graph, indicating a significant change in the noise characteristics as a result of the graph convolution process. We provide a demonstrative visualization in Figure 6.*

$\square$

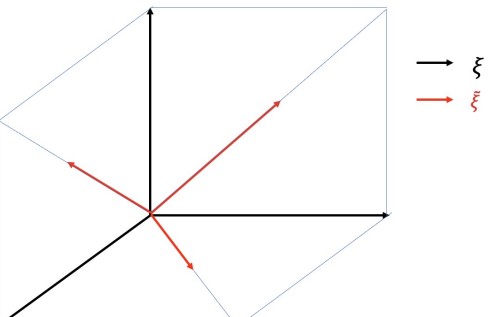

Figure 6: An illustrative example of noise vector before and after graph aggregation. In this example, we consider $d = 3$ and all degree are 1. The black vectors stand for noise vectors $\boldsymbol{\xi}$ before graph convolution. Each of them are orthogonal to each other. The red vectors represent noise vectors after graph convolution $\tilde{\boldsymbol{\xi}}$. They are graph convoluted noise vectors of two original noise vectors. Note that the $\ell_2$ norm between two kinds of vector follows $\|\tilde{\boldsymbol{\xi}}\|_2 = \frac{\sqrt{2}}{2}\|\boldsymbol{\xi}\|_2$. This plot demonstrates how graph convolution shrinks the $\ell_2$ norm of noise vectors.

**Lemma A.7.** *Suppose that $d = \Omega(n(p+s)\log(nm/\delta))$, $m = \Omega(\log(1/\delta))$. Then with probability at least $1 - \delta$,*

$$|\langle \mathbf{w}_{j,r}^{(0)}, \tilde{\boldsymbol{\xi}}_i \rangle| \le 4\sqrt{\log(8mn/\delta)} \cdot \sigma_0 \sigma_p \sqrt{d/(n(p+s))},$$

$$\sigma_0 \sigma_p \sqrt{d/(n(p+s))}/4 \le \max_{r \in [m]} j \cdot \langle \mathbf{w}_{j,r}^{(0)}, \tilde{\boldsymbol{\xi}}_i \rangle \le 2\sqrt{\log(8mn/\delta)} \cdot \sigma_0 \sigma_p \sqrt{d/(n(p+s))},$$

*for all $j \in \{\pm 1\}$ and $i \in [n]$.*

*Proof of Lemma A.7.* According to the fact that the weight $\mathbf{w}_{j,r}(0)$ and noise vector $\boldsymbol{\xi}$ are sampled from Gaussian distribution, we know that $\langle \mathbf{w}_{j,r}^{(0)}, \tilde{\boldsymbol{\xi}}_i \rangle$ is also Gaussian. By Lemma A.5, with probability at least $1 - \delta/4$, we have that

$$\sigma_p \sqrt{d/(n(p+s))}/\sqrt{2} \le \|\tilde{\boldsymbol{\xi}}_i\|_2 \le \sqrt{3/2} \cdot \sigma_p \sqrt{d/(n(p+s))}$$

holds for all $i \in [n]$. Therefore, applying the concentration bound for Gaussian variable, we obtain that

$$|\langle \mathbf{w}_{j,r}^{(0)}, \tilde{\boldsymbol{\xi}}_i \rangle| \le 4\sqrt{\log(8mn/\delta)} \cdot \sigma_0 \sigma_p \sqrt{d/(n(p+s))}.$$

Next we finish the argument for the lower bound of maximum through the follow expression:

$$P(\max\langle \mathbf{w}_{j,r}^{(0)}, \tilde{\boldsymbol{\xi}}_i \rangle \ge \sigma_0 \sigma_p \sqrt{d/(n(p+s))}/4) = 1 - P(\max\langle \mathbf{w}_{j,r}^{(0)}, \tilde{\boldsymbol{\xi}}_i \rangle < \sigma_0 \sigma_p \sqrt{d/(n(p+s))}/4)$$

$$= 1 - P(\max\langle \mathbf{w}_{j,r}^{(0)}, \tilde{\boldsymbol{\xi}}_i \rangle < \sigma_0 \sigma_p \sqrt{d/(n(p+s))}/4)^{2m}$$

$$\ge 1 - \delta/4.$$

Together with Lemma A.5, we finally obtain that

$$\sigma_0 \sigma_p \sqrt{d/(n(p+s))}/4 \le \max_{r \in [m]} j \cdot \langle \mathbf{w}_{j,r}^{(0)}, \tilde{\boldsymbol{\xi}}_i \rangle \le 2\sqrt{\log(8mn/\delta)} \cdot \sigma_0 \sigma_p \sqrt{d/(n(p+s))}.$$

$\square$

# B GENERAL LEMMAS FOR ITERATIVE COEFFICIENT ANALYSIS

In this section, we deliver lemmas that delineate the iterative behavior of coefficients under gradient descent. We commence with proving the coefficient update rules as stated in Lemma 5.1 in Section B.1. Subsequently, we establish the scale of training dynamics in Section B.2.

## B.1 COEFFICIENT UPDATE RULE

**Lemma B.1** (Restatement of Lemma 5.1). *The coefficients $\gamma_{j,r}^{(t)}, \overline{\rho}_{j,r,i}^{(t)}, \underline{\rho}_{j,r,i}^{(t)}$ defined in Eq. (9) satisfy the following iterative equations:*

$$\gamma_{j,r}^{(0)}, \overline{\rho}_{j,r,i}^{(0)}, \underline{\rho}_{j,r,i}^{(0)} = 0,$$

$$\gamma_{j,r}^{(t+1)} = \gamma_{j,r}^{(t)} - \frac{\eta}{nm} \cdot \sum_{i=1}^{n} \ell_i'^{(t)} \sigma'(\langle \mathbf{w}_{j,r}^{(t)}, \tilde{y}_i \boldsymbol{\mu} \rangle) y_i \tilde{y}_i \|\boldsymbol{\mu}\|_2^2,$$

$$\overline{\rho}_{j,r,i}^{(t+1)} = \overline{\rho}_{j,r,i}^{(t)} - \frac{\eta}{nm} \cdot \sum_{k \in \mathcal{N}(i)} D_k^{-1} \cdot \ell_k'^{(t)} \cdot \sigma'(\langle \mathbf{w}_{j,r}^{(t)}, \tilde{\boldsymbol{\xi}}_k \rangle) \cdot \|\boldsymbol{\xi}_i\|_2^2 \cdot \mathbb{1}(y_k = j),$$

$$\underline{\rho}_{j,r,i}^{(t+1)} = \underline{\rho}_{j,r,i}^{(t)} - \frac{\eta}{nm} \cdot \sum_{k \in \mathcal{N}(i)} D_k^{-1} \cdot \ell_k'^{(t)} \cdot \sigma'(\langle \mathbf{w}_{j,r}^{(t)}, \tilde{\boldsymbol{\xi}}_k \rangle) \cdot \|\boldsymbol{\xi}_i\|_2^2 \cdot \mathbb{1}(y_k = -j),$$

*for all $r \in [m]$, $j \in \{\pm 1\}$ and $i \in [n]$.*

**Remark B.2.** *This lemma serves as a foundational element in our analysis of dynamics. Initially, the study of neural network dynamics under gradient descent required us to monitor the fluctuations in weights. However, this Lemma enables us to observe these dynamics through a new lens, focusing on two distinct aspects: signal learning and noise memorization. These are represented by the*

variables $\gamma_{j,r}^{(t)}$ and $\rho_{j,r,i}^{(t)}$, respectively. Furthermore, the selection of our data model was a conscious decision, designed to clearly separate the signal learning from the noise memorization aspects of learning. By maintaining a clear distinction between signal and noise, we can conduct a precise analysis of how each model learns the signal and memorizes the noise. This approach not only simplifies our understanding but also enhances our ability to dissect the underlying mechanisms of learning.

*Proof of Lemma B.1.* Basically, the iteration of coefficients is derived based on gradient descent rule equation 7 and weight decomposition equation 9. We first consider $\hat{\gamma}_{j,r}^{(0)}, \hat{\rho}_{j,r,i}^{(0)} = 0$ and

$$\hat{\gamma}_{j,r}^{(t+1)} = \hat{\gamma}_{j,r}^{(t)} - \frac{\eta}{nm} \cdot \sum_{i=1}^{n} \ell_i'^{(t)} \sigma'(\langle \mathbf{w}_{j,r}^{(t)}, \tilde{y}_i \boldsymbol{\mu}_i \rangle) y_i \tilde{y}_i \|\boldsymbol{\mu}\|_2^2,$$

$$\hat{\rho}_{j,r,i}^{(t+1)} = \hat{\rho}_{j,r,i}^{(t)} - \frac{\eta}{nm} \cdot \sum_{k \in \mathcal{N}(i)} D_k^{-1} \cdot \ell_k'^{(t)} \cdot \sigma'(\langle \mathbf{w}_{j,r}^{(t)}, \tilde{\boldsymbol{\xi}}_k \rangle) \cdot \|\boldsymbol{\xi}_i\|_2^2 \cdot y_k,$$

Taking above equations into Equation equation 7, we can obtain that

$$\mathbf{w}_{j,r}^{(t)} = \mathbf{w}_{j,r}^{(0)} + j \cdot \hat{\gamma}_{j,r}^{(t)} \cdot \|\boldsymbol{\mu}\|_2^{-2} \cdot \boldsymbol{\mu} + \sum_{i=1}^{n} \hat{\rho}_{j,r,i}^{(t)} \|\boldsymbol{\xi}_i\|_2^{-2} \cdot \boldsymbol{\xi}_i.$$

This result verifies that the iterative update of the coefficients is directly driven by the gradient descent update process. Furthermore, the uniqueness of the decomposition leads us to the precise relationships $\gamma_{j,r}^{(t)} = \hat{\gamma}_{j,r}^{(t)}$ and $\rho_{j,r,i}^{(t)} = \hat{\rho}_{j,r,i}^{(t)}$. Next, we examine the stability of the sign associated with noise memorization by employing the following telescopic analysis. This method allows us to investigate the continuity and consistency of the noise memorization process, providing insights into how the system behaves over successive iterations.

$$\rho_{j,r,i}^{(t)} = -\sum_{s=0}^{t-1} \sum_{k \in \mathcal{N}(i)} D_k^{-1} \frac{\eta}{nm} \cdot \ell_k'^{(s)} \cdot \sigma'(\langle \mathbf{w}_{j,r}^{(s)}, \tilde{\boldsymbol{\xi}}_k \rangle) \cdot \|\boldsymbol{\xi}_i\|_2^2 \cdot jy_k.$$

Recall the sign of loss derivative is given by the definition of the cross-entropy loss, namely, $\ell_i'^{(t)} < 0$. Therefore,

$$\overline{\rho}_{j,r,i}^{(t)} = -\sum_{s=0}^{t-1} \frac{\eta}{nm} \cdot \sum_{k \in \mathcal{N}(i)} D_k^{-1} \cdot \ell_k'^{(t)} \cdot \sigma'(\langle \mathbf{w}_{j,r}^{(t)}, \tilde{\boldsymbol{\xi}}_k \rangle) \cdot \|\boldsymbol{\xi}_i\|_2^2 \cdot \mathbb{1}(y_k = j), \tag{14}$$

$$\underline{\rho}_{j,r,i}^{(t)} = -\sum_{s=0}^{t-1} \frac{\eta}{nm} \cdot \sum_{k \in \mathcal{N}(i)} D_k^{-1} \cdot \ell_k'^{(t)} \cdot \sigma'(\langle \mathbf{w}_{j,r}^{(t)}, \tilde{\boldsymbol{\xi}}_k \rangle) \cdot \|\boldsymbol{\xi}_i\|_2^2 \cdot \mathbb{1}(y_k = -j). \tag{15}$$

Writing out the iterative versions of equation 14 and equation 15 completes the proof. □

**Remark B.3.** *The proof strategy follows the study of feature learning in CNN as described in (Cao et al., 2022). However, compared to CNNs, the decomposition of weights in GNN is notably more intricate. This complexity is particularly evident in the dynamics of noise memorization, as represented by Equations 14) and 15). The reason for this increased complexity lies in the additional graph convolution operations within GNNs. These operations introduce new interaction and dependencies, making the analysis of weight dynamics more challenging and nuanced.*

## B.2 SCALE OF TRAINING DYNAMICS

Our proof hinges on a meticulous evaluation of the coefficient values $\gamma_{j,r}^{(t)}, \overline{\rho}_{j,r,i}^{(t)}, \underline{\rho}_{j,r,i}^{(t)}$ throughout the entire training process. In order to facilitate a more thorough analysis, we first establish the following bounds for these coefficients, which are maintained consistently throughout the training period.

Consider training the Graph Neural Network (GNN) for an extended period up to $T^*$. We aim to investigate the scale of noise memorization in relation to signal learning.

Let $T^* = \eta^{-1}\mathrm{poly}(\epsilon^{-1}, \|\boldsymbol{\mu}\|_2^{-1}, d^{-1}\sigma_p^{-2}, \sigma_0^{-1}, n, m, d)$ be the maximum admissible iterations. Denote $\alpha = 4\log(T^*)$. In preparation for an in-depth analysis, we enumerate the necessary conditions that must be satisfied. These conditions, which are essential for the subsequent examination, are also detailed in Condition 4.1:

$$\eta = O\Big(\min\{nm/(q\sigma_p^2 d), nm/(q2^{q+2}\alpha^{q-2}\sigma_p^2 d), nm/(q2^{q+2}\alpha^{q-2}\|\boldsymbol{\mu}\|_2^2)\}\Big), \tag{16}$$

$$\sigma_0 \le [16\sqrt{\log(8mn/\delta)}]^{-1}\min\left\{\Xi^{-1}\|\boldsymbol{\mu}\|_2^{-1}, (\sigma_p\sqrt{d/(n(p+s))})^{-1}\right\}, \tag{17}$$

$$d \ge 1024\log(4n^2/\delta)\alpha^2 n^2. \tag{18}$$

Denote $\beta = 2\max_{i,j,r}\{|\langle \mathbf{w}_{j,r}^{(0)}, \tilde{y}_i \cdot \boldsymbol{\mu}\rangle|, |\langle \mathbf{w}_{j,r}^{(0)}, \tilde{\boldsymbol{\xi}}_i\rangle|\}$, it is straightforward to show the following inequality:

$$4\max\left\{\beta, 8n\sqrt{\frac{\log(4n^2/\delta)}{d}}\alpha\right\} \le 1. \tag{19}$$

First, by Lemma A.4 with probability at least $1 - \delta$, we can upper bound $\beta$ by $4\sqrt{\log(8mn/\delta)}\cdot\sigma_0\cdot\max\{\Xi\|\boldsymbol{\mu}\|_2, \sigma_p\sqrt{d/(n(p+s))}\}$. Combined with the condition equation 17, we can bound $\beta$ by 1. Second, it is easy to check that $8n\sqrt{\frac{\log(4n^2/\delta)}{d}}\alpha \le 1$ by inequality equation 18.

Having established the values of $\alpha$ and $\beta$ at hand, we are now in a position to assert that the following proposition holds for the entire duration of the training process, specifically for $0 \le t \le T^*$.

**Proposition B.4.** *Under Condition 4.1, for $0 \le t \le T^*$, where $T^* = \eta^{-1}\mathrm{poly}(\epsilon^{-1}, \|\boldsymbol{\mu}\|_2^{-1}, d^{-1}\sigma_p^{-2}, \sigma_0^{-1}, n, m, d)$, we have that*

$$0 \le \gamma_{j,r}^{(t)}, \overline{\rho}_{j,r,i}^{(t)} \le \alpha, \tag{20}$$

$$0 \ge \underline{\rho}_{j,r,i}^{(t)} \ge -\alpha, \tag{21}$$

*for all $r \in [m]$, $j \in \{\pm 1\}$ and $i \in [n]$, where $\alpha = 4\log(T^*)$.*

To establish Proposition B.4, we will employ an inductive approach. Before proceeding with the proof, we need to introduce several technical lemmas that are fundamental to our argument.

We note that although the setting is slightly different from the case in (Cao et al., 2022). With the same analysis, we can obtain the following result.

**Lemma B.5** ((Cao et al., 2022)). *For any $t \ge 0$, it holds that $\langle \mathbf{w}_{j,r}^{(t)} - \mathbf{w}_{j,r}^{(0)}, \boldsymbol{\mu}\rangle = j \cdot \gamma_{j,r}^{(t)}$ for all $r \in [m]$, $j \in \{\pm 1\}$.*

In the subsequent three lemmas, our proof strategy is guided by the approach found in (Cao et al., 2022). However, we extend this methodology by providing a fine-grained analysis that takes into account the additional complexity introduced by the graph convolution operation.

**Lemma B.6.** *Under Condition 4.1, suppose equation 20 and equation 21 hold at iteration $t$. Then*

$$\hat{\rho}_{j,r,i}^{(t)} - 8n\sqrt{\frac{\log(4n^2/\delta)}{d}}\alpha \le \langle \mathbf{w}_{j,r}^{(t)} - \mathbf{w}_{j,r}^{(0)}, \tilde{\boldsymbol{\xi}}_i\rangle \le \hat{\rho}_{j,r,i}^{(t)} + 8n\sqrt{\frac{\log(4n^2/\delta)}{d}}\alpha,$$

*where $\hat{\rho}_{j,r,i}^{(t)} \triangleq \sum_{k\in\mathcal{N}(i)} D_i^{-1}\sum_{i'\ne k}\rho_{j,r,i'}^{(t)}$, for all $r \in [m]$, $j \in \{\pm 1\}$ and $i \in [n]$.*

**Remark B.7.** *Lemma B.6 asserts that the inner product between the updated weight and the graph convolution operation closely approximates the graph-convoluted noise memorization.*

*Proof of Lemma B.6.* It is known that,

$$
\begin{aligned}
\langle \mathbf{w}_{j,r}^{(t)} - \mathbf{w}_{j,r}^{(0)}, \tilde{\boldsymbol{\xi}}_i \rangle &= \sum_{i'=1}^{n} \overline{\rho}_{j,r,i'}^{(t)} \|\boldsymbol{\xi}_{i'}\|_2^{-2} \cdot \langle \boldsymbol{\xi}_{i'}, \tilde{\boldsymbol{\xi}}_i \rangle + \sum_{i'=1}^{n} \underline{\rho}_{j,r,i'}^{(t)} \|\boldsymbol{\xi}_{i'}\|_2^{-2} \cdot \langle \boldsymbol{\xi}_{i'}, \tilde{\boldsymbol{\xi}}_i \rangle \\
&= \sum_{i'=1}^{n} \sum_{k \in \mathcal{N}(i)} D_i^{-1} \overline{\rho}_{j,r,i'}^{(t)} \|\boldsymbol{\xi}_{i'}\|_2^{-2} \cdot \langle \boldsymbol{\xi}_{i'}, \boldsymbol{\xi}_k \rangle + \sum_{i'=1}^{n} \sum_{k \in \mathcal{N}(i)} D_i^{-1} \underline{\rho}_{j,r,i'}^{(t)} \|\boldsymbol{\xi}_{i'}\|_2^{-2} \cdot \langle \boldsymbol{\xi}_{i'}, \boldsymbol{\xi}_k \rangle \\
&\leq 4\sqrt{\frac{\log(4n^2/\delta)}{d}} \sum_{k \in \mathcal{N}(i)} D_i^{-1} \sum_{i' \neq k} |\overline{\rho}_{j,r,i'}^{(t)}| + 4\sqrt{\frac{\log(4n^2/\delta)}{d}} \sum_{k \in \mathcal{N}(i)} D_i^{-1} \sum_{i' \neq k} |\underline{\rho}_{j,r,i'}^{(t)}| \\
&\quad + \sum_{k \in \mathcal{N}(i)} D_i^{-1} \sum_{i' \neq k} \overline{\rho}_{j,r,i'}^{(t)} + \sum_{k \in \mathcal{N}(i)} D_i^{-1} \sum_{i' \neq k} \underline{\rho}_{j,r,i'}^{(t)} \\
&\leq \hat{\rho}_{j,r,i}^{(t)} + 8n\sqrt{\frac{\log(4n^2/\delta)}{d}} \alpha,
\end{aligned}
$$

where we define $\hat{\rho}_{j,r,i} \triangleq \sum_{k \in \mathcal{N}(i)} D_i^{-1} \sum_{i' \neq k} \rho_{j,r,i'}^{(t)}$ the second inequality is by Lemma A.1 and the last inequality is by $|\overline{\rho}_{j,r,i'}^{(t)}|, |\underline{\rho}_{j,r,i'}^{(t)}| \leq \alpha$ in equation 20.

Similarly, we can show that:

$$
\begin{aligned}
\langle \mathbf{w}_{j,r}^{(t)} - \mathbf{w}_{j,r}^{(0)}, \tilde{\boldsymbol{\xi}}_i \rangle &= \sum_{i'=1}^{n} \overline{\rho}_{j,r,i'}^{(t)} \|\boldsymbol{\xi}_{i'}\|_2^{-2} \cdot \langle \boldsymbol{\xi}_{i'}, \tilde{\boldsymbol{\xi}}_i \rangle + \sum_{i'=1}^{n} \underline{\rho}_{j,r,i'}^{(t)} \|\boldsymbol{\xi}_{i'}\|_2^{-2} \cdot \langle \boldsymbol{\xi}_{i'}, \tilde{\boldsymbol{\xi}}_i \rangle \\
&= \sum_{i'=1}^{n} \sum_{k \in \mathcal{N}(i)} D_i^{-1} \overline{\rho}_{j,r,i'}^{(t)} \|\boldsymbol{\xi}_{i'}\|_2^{-2} \cdot \langle \boldsymbol{\xi}_{i'}, \boldsymbol{\xi}_k \rangle + \sum_{i'=1}^{n} \sum_{k \in \mathcal{N}(i)} D_i^{-1} \underline{\rho}_{j,r,i'}^{(t)} \|\boldsymbol{\xi}_{i'}\|_2^{-2} \cdot \langle \boldsymbol{\xi}_{i'}, \boldsymbol{\xi}_k \rangle \\
&\geq -4\sqrt{\frac{\log(4n^2/\delta)}{d}} \sum_{k \in \mathcal{N}(i)} D_i^{-1} \sum_{i' \neq k} |\overline{\rho}_{j,r,i'}^{(t)}| - 4\sqrt{\frac{\log(4n^2/\delta)}{d}} \sum_{k \in \mathcal{N}(i)} D_i^{-1} \sum_{i' \neq k} |\underline{\rho}_{j,r,i'}^{(t)}| \\
&\quad + \sum_{k \in \mathcal{N}(i)} D_i^{-1} \sum_{i' \neq k} \overline{\rho}_{j,r,i'}^{(t)} + \sum_{k \in \mathcal{N}(i)} D_i^{-1} \sum_{i' \neq k} \underline{\rho}_{j,r,i'}^{(t)} \\
&\geq \hat{\rho}_{j,r,i}^{(t)} - 8n\sqrt{\frac{\log(4n^2/\delta)}{d}} \alpha,
\end{aligned}
$$

where the first inequality is by Lemma A.1 and the second inequality is by $|\overline{\rho}_{j,r,i'}^{(t)}|, |\underline{\rho}_{j,r,i'}^{(t)}| \leq \alpha$ in equation 20, which completes the proof. $\square$

**Lemma B.8.** *Under Condition 4.1, suppose equation 20 and equation 21 hold at iteration t. Then*

$$
\langle \mathbf{w}_{j,r}^{(t)}, \tilde{y}_i \boldsymbol{\mu} \rangle \leq \langle \mathbf{w}_{j,r}^{(0)}, \tilde{y}_i \boldsymbol{\mu} \rangle,
$$

$$
\langle \mathbf{w}_{j,r}^{(t)}, \tilde{\boldsymbol{\xi}}_i \rangle \leq \langle \mathbf{w}_{j,r}^{(0)}, \tilde{\boldsymbol{\xi}}_i \rangle + 8n\sqrt{\frac{\log(4n^2/\delta)}{d}} \alpha,
$$

*for all $r \in [m]$ and $j \neq y_i$. If $\max\{\gamma_{j,r}^{(t)}, \rho_{j,r,i}^{(t)}\} = O(1)$, we further have that $F_j(\mathbf{W}_j^{(t)}, \tilde{\mathbf{x}}_i) = O(1)$.*

**Remark B.9.** *Lemma B.8 further establishes that the update in the direction of $\tilde{\boldsymbol{\xi}}$ can be constrained within specific bounds when $j \neq y_i$. As a result, the output function remains controlled and does not exceed a constant order.*

*Proof of Lemma B.8.* For $j \neq y_i$, we have that

$$
\langle \mathbf{w}_{j,r}^{(t)}, \tilde{y}_i \boldsymbol{\mu} \rangle = \langle \mathbf{w}_{j,r}^{(0)}, \tilde{y}_i \boldsymbol{\mu} \rangle + \tilde{y}_i \cdot j \cdot \gamma_{j,r}^{(t)} \leq \langle \mathbf{w}_{j,r}^{(0)}, \tilde{y}_i \boldsymbol{\mu} \rangle, \tag{22}
$$

where the inequality is by $\gamma_{j,r}^{(t)} \geq 0$ and Lemma A.4 stating that $\text{sign}(y_i) = \text{sign}(\tilde{y}_i)$ with a high probability. In addition, we have

$$
\begin{aligned}
\langle \mathbf{w}_{j,r}^{(t)}, \tilde{\boldsymbol{\xi}}_i \rangle &= \langle \mathbf{w}_{j,r}^{(0)}, \tilde{\boldsymbol{\xi}}_i \rangle + \sum_{k \in \mathcal{N}(i)} D_i^{-1} \sum_{i'=1}^{n} \rho_{j,r,i'} \langle \boldsymbol{\xi}_k, \boldsymbol{\xi}_{i'} \rangle \|\boldsymbol{\xi}_{i'}\|_2^{-2} \\
&\leq \langle \mathbf{w}_{j,r}^{(0)}, \tilde{\boldsymbol{\xi}}_i \rangle + D_i^{-1} \left( \sum_{y_k \neq j} \underline{\rho}_{j,r,i}^{(t)} + \sum_{y_k = j} \overline{\rho}_{j,r,i}^{(t)} \right) + 8n\sqrt{\frac{\log(4n^2/\delta)}{d}} \alpha \\
&\leq \langle \mathbf{w}_{j,r}^{(0)}, \tilde{\boldsymbol{\xi}}_i \rangle + 8n\sqrt{\frac{\log(4n^2/\delta)}{d}} \alpha,
\end{aligned}
\tag{23}
$$

where the first inequality is by Lemma B.6 and the second inequality is due to $\hat{\rho}_{j,r,i}^{(t)} \leq 0$ based on Lemma A.4. Then we can get that

$$
\begin{aligned}
F_j(\mathbf{W}_j^{(t)}, \tilde{\mathbf{x}}_i) &= \frac{1}{m} \sum_{r=1}^{m} [\sigma(\langle \mathbf{w}_{j,r}^{(t)}, \tilde{y}_i \cdot \boldsymbol{\mu} \rangle) + \sigma(\langle \mathbf{w}_{j,r}^{(t)}, \tilde{\boldsymbol{\xi}}_i \rangle)] \\
&= \frac{1}{m} \sum_{r=1}^{m} [\sigma(\langle \mathbf{w}_{j,r}^{(t)}, \tilde{y}_i \cdot \boldsymbol{\mu} \rangle) + \sigma(\langle \mathbf{w}_{j,r}^{(t)}, D_i^{-1} \sum_{k \in \mathcal{N}(i)} \boldsymbol{\xi}_k \rangle)] \\
&= \frac{1}{m} \sum_{r=1}^{m} [\sigma(\langle \mathbf{w}_{j,r}^{(0)}, \tilde{y}_i \cdot \boldsymbol{\mu} \rangle) + \sigma(\langle \mathbf{w}_{j,r}^{(0)}, \tilde{\boldsymbol{\xi}}_i \rangle + \langle \mathbf{w}_{j,r}^{(t)} - \mathbf{w}_{j,r}^{(0)}, D_i^{-1} \sum_{k \in \mathcal{N}(i)} \boldsymbol{\xi}_k \rangle)] \\
&\leq \frac{1}{m} \sum_{r=1}^{m} [\sigma(\langle \mathbf{w}_{j,r}^{(0)}, \tilde{y}_i \cdot \boldsymbol{\mu} \rangle) + \sigma(\langle \mathbf{w}_{j,r}^{(0)}, \tilde{\boldsymbol{\xi}}_i \rangle + 8n\sqrt{\frac{\log(4n^2/\delta)}{d}} \alpha + \hat{\rho}_{j,r,i}^{(t)})] \\
&\leq 2^{q+1} \max_{j,r,i} \left\{ |\langle \mathbf{w}_{j,r}^{(0)}, \tilde{y}_i \cdot \boldsymbol{\mu} \rangle|, |\langle \mathbf{w}_{j,r}^{(0)}, \tilde{\boldsymbol{\xi}}_i \rangle|, 8n\sqrt{\frac{\log(4n^2/\delta)}{d}} \alpha \right\}^q \\
&\leq 1,
\end{aligned}
$$

where the first inequality is by equation 22, equation 23 and the second inequality is by equation 19 and $\max\{\gamma_{j,r}^{(t)}, \rho_{j,r,i}^{(t)}\} = O(1)$. $\qquad \square$

**Lemma B.10.** *Under Condition 4.1, suppose equation 20 and equation 21 hold at iteration $t$. Then*

$$
\langle \mathbf{w}_{j,r}^{(t)}, \tilde{y}_i \boldsymbol{\mu} \rangle = \langle \mathbf{w}_{j,r}^{(0)}, \tilde{y}_i \boldsymbol{\mu} \rangle + \gamma_{j,r}^{(t)},
$$

$$
\langle \mathbf{w}_{j,r}^{(t)}, \tilde{\boldsymbol{\xi}}_i \rangle \leq \langle \mathbf{w}_{j,r}^{(0)}, \tilde{\boldsymbol{\xi}}_i \rangle + \hat{\rho}_{j,r,i}^{(t)} + 8n\sqrt{\frac{\log(4n^2/\delta)}{d}} \alpha
$$

*for all $r \in [m]$, $j = y_i$ and $i \in [n]$. If $\max\{\gamma_{j,r}^{(t)}, \rho_{j,r,i}^{(t)}\} = O(1)$, we further have that $F_j(\mathbf{W}_j^{(t)}, \tilde{\mathbf{x}}_i) = O(1)$.*

**Remark B.11.** *Lemma B.10 further establishes that the update in the direction of $\boldsymbol{\mu}$ and $\tilde{\boldsymbol{\xi}}$ can be constrained within specific bounds when $j = y_i$. As a result, the output function remains controlled and does not exceed a constant order with an additional condition.*

*Proof of Lemma B.10.* For $j = y_i$, we have that

$$
\langle \mathbf{w}_{j,r}^{(t)}, \tilde{y}_i \boldsymbol{\mu} \rangle = \langle \mathbf{w}_{j,r}^{(0)}, \tilde{y}_i \boldsymbol{\mu} \rangle + \gamma_{j,r}^{(t)},
\tag{24}
$$

where the equation is by Lemma B.5. We also have that

$$
\langle \mathbf{w}_{j,r}^{(t)}, \tilde{\boldsymbol{\xi}}_i \rangle \leq \langle \mathbf{w}_{j,r}^{(0)}, \tilde{\boldsymbol{\xi}}_i \rangle + \hat{\rho}_{j,r,i}^{(t)} + 8n\sqrt{\frac{\log(4n^2/\delta)}{d}} \alpha,
\tag{25}
$$

where the inequality is by Lemma B.6. If $\max\{\gamma_{j,r}^{(t)}, \rho_{j,r,i}^{(t)}\} = O(1)$, we have following bound

$$
\begin{aligned}
F_j(\mathbf{W}_j^{(t)}, \tilde{\mathbf{x}}_i) &= \frac{1}{m} \sum_{r=1}^{m} [\sigma(\langle \mathbf{w}_{j,r}^{(t)}, \tilde{y}_i \cdot \boldsymbol{\mu} \rangle) + \sigma(\langle \mathbf{w}_{j,r}^{(t)}, \tilde{\boldsymbol{\xi}}_i \rangle)] \\
&\leq 2 \cdot 3^q \max_{j,r,i} \left\{ \gamma_{j,r}^{(t)}, |\hat{\rho}_{j,r,i}^{(t)}|, |\langle \mathbf{w}_{j,r}^{(0)}, \tilde{y}_i \cdot \boldsymbol{\mu} \rangle\rangle|, |\langle \mathbf{w}_{j,r}^{(0)}, \tilde{\boldsymbol{\xi}}_i \rangle|, 8n\sqrt{\frac{\log(4n^2/\delta)}{d}} \alpha \right\}^q \\
&= O(1),
\end{aligned}
$$

where $\hat{\rho}_{j,r,i}^{(t)} = \frac{1}{D_i} \sum_{k \in \mathcal{N}(i)} \overline{\rho}_{j,r,k}^{(t)} \mathbb{1}(y_k = j) + \underline{\rho}_{j,r,k}^{(t)} \mathbb{1}(y_k \neq j)$, the first inequality is by equation 24, equation 25. Then the second inequality is by equation 19 where $\beta = 2 \max_{i,j,r}\{|\langle \mathbf{w}_{j,r}^{(0)}, \tilde{y}_i \cdot \boldsymbol{\mu} \rangle|, |\langle \mathbf{w}_{j,r}^{(0)}, \tilde{\boldsymbol{\xi}}_i \rangle|\} \leq 1$ and condition that $\max\{\gamma_{j,r}^{(t)}, \rho_{j,r,i}^{(t)}\} = O(1)$. $\qquad \square$

Equipped with Lemmas B.5 - B.10, we are now prepared to prove Proposition B.4. These lemmas provide the foundational building blocks and insights necessary for our proof, setting the stage for a rigorous and comprehensive demonstration of the proposition

*Proof of Proposition B.4.* Following a similar approach to the proof found in (Cao et al., 2022), we employ an induction method. This technique allows us to build our argument step by step, drawing on established principles and extending them to our specific context, thereby providing a robust and systematic demonstration.

At the initial time step $t = 0$, the outcome is clear since all coefficients are set to zero.

Next, we hypothesize that there exists a time $\tilde{T}$ less that $T^*$ during which Proposition B.4 holds true for every moment within the range $0 \leq t \leq \tilde{T} - 1$. Our objective is to show that this proposition remains valid at $t = \tilde{T}$.

We aim to validate that equation equation 21 is applicable at $t = \tilde{T}$, meaning that,

$$
\underline{\rho}_{j,r,i}^{(t)} \geq -\beta - 16n\sqrt{\frac{\log(4n^2/\delta)}{d}} \alpha,
$$

for the given parameters. It's important to note that $\underline{\rho}_{j,r,i}^{(t)} = 0$ when $j = y_i$. So we only need to consider instances where $j \neq y_i$.

1) Under condition

$$
\underline{\rho}_{j,r,i}^{(\tilde{T}-1)} \leq -0.5\beta - 8n\sqrt{\frac{\log(4n^2/\delta)}{d}} \alpha,
$$

Lemma B.6 leads us to the following relationships:

$$
\langle \mathbf{w}_{j,r}^{(\tilde{T}-1)}, \tilde{y}_i \boldsymbol{\mu} \rangle \leq \hat{\rho}_{j,r,i}^{(\tilde{T}-1)} + \langle \mathbf{w}_{j,r}^{(0)}, \tilde{y}_i \boldsymbol{\mu} \rangle + 8n\sqrt{\frac{\log(4n^2/\delta)}{d}} \alpha \leq 0,
$$

and thus

$$
\begin{aligned}
\underline{\rho}_{j,r,i}^{(\tilde{T})} &= \underline{\rho}_{j,r,i}^{(\tilde{T}-1)} + \frac{\eta}{nm} \sum_k D_k^{-1} \cdot \ell_k'^{(\tilde{T}-1)} \cdot \sigma'(\langle \mathbf{w}_{j,r}^{(\tilde{T}-1)}, \tilde{\boldsymbol{\xi}}_k \rangle) \cdot \mathbb{1}(y_k = -j) \|\boldsymbol{\xi}_i\|_2^2 \\
&= \underline{\rho}_{j,r,i}^{(\tilde{T}-1)} \geq -\beta - 16n\sqrt{\frac{\log(4n^2/\delta)}{d}} \alpha,
\end{aligned}
$$

with the final inequality being supported by the induction hypothesis.

2) Given the condition $\underline{\rho}_{j,r,i}^{(\tilde{T}-1)} \geq -0.5\beta - 8n\sqrt{\frac{\log(4n^2/\delta)}{d}}\alpha$, we can derive the following:

$$
\begin{aligned}
\underline{\rho}_{j,r,i}^{(\tilde{T})} &= \underline{\rho}_{j,r,i}^{(\tilde{T}-1)} + \frac{\eta}{nm} \cdot \sum_{k \in \mathcal{N}(i)} D_k^{-1} \ell_k'^{(\tilde{T}-1)} \cdot \sigma'(\langle \mathbf{w}_{j,r}^{(T-1)}, \tilde{\boldsymbol{\xi}}_k \rangle) \cdot \mathbb{1}(y_k = -j)\|\boldsymbol{\xi}_i\|_2^2 \\
&\geq -0.5\beta - 8n\sqrt{\frac{\log(4n^2/\delta)}{d}}\alpha - O\left(\frac{\eta\sigma_p^2 d}{nm}\right)\sigma'\left(0.5\beta + 8n\sqrt{\frac{\log(4n^2/\delta)}{d}}\alpha\right) \\
&\geq -0.5\beta - 8n\sqrt{\frac{\log(4n^2/\delta)}{d}}\alpha - O\left(\frac{\eta q\sigma_p^2 d}{nm}\right)\left(0.5\beta + 8n\sqrt{\frac{\log(4n^2/\delta)}{d}}\alpha\right) \\
&\geq -\beta - 16n\sqrt{\frac{\log(4n^2/\delta)}{d}}\alpha,
\end{aligned}
$$

where we apply the inequalities $\ell_i'^{(\tilde{T}-1)} \leq 1$ and $\|\boldsymbol{\xi}_i\|_2 = O(\sigma_p^2 d)$, and use the conditions $\eta = O(nm/(q\sigma_p^2 d))$ and $0.5\beta + 8n\sqrt{\frac{\log(4n^2/\delta)}{d}}\alpha \leq 1$, as specified in equation 16.

Next, we aim to show that equation 20 is valid for $t = \tilde{T}$. We can express:

$$
\begin{aligned}
|\ell_i'^{(t)}| &= \frac{1}{1 + \exp\{y_i \cdot [F_{+1}(\mathbf{W}_{+1}^{(t)}, \tilde{\mathbf{x}}_i) - F_{-1}(\mathbf{W}_{-1}^{(t)}, \tilde{\mathbf{x}}_i)]\}} \\
&\leq \exp\{-y_i \cdot [F_{+1}(\mathbf{W}_{+1}^{(t)}, \tilde{\mathbf{x}}_i) - F_{-1}(\mathbf{W}_{-1}^{(t)}, \tilde{\mathbf{x}}_i)]\} \\
&\leq \exp\{-F_{y_i}(\mathbf{W}_{y_i}^{(t)}, \tilde{\mathbf{x}}_i) + 1\}.
\end{aligned}
\tag{26}
$$

with the last inequality being a result of Lemma B.8. Additionally, we recall the update rules for $\gamma_{j,r}^{(t+1)}$ and $\overline{\rho}_{j,r,i}^{(t+1)}$:

$$
\begin{aligned}
\gamma_{j,r}^{(t+1)} &= \gamma_{j,r}^{(t)} - \frac{\eta}{nm} \cdot \sum_{i=1}^{n} \ell_i'^{(t)} \cdot \sigma'(\langle \mathbf{w}_{j,r}^{(t)}, \tilde{y}_i \cdot \boldsymbol{\mu}\rangle) y_i \tilde{y}_i \|\boldsymbol{\mu}\|_2^2, \\
\overline{\rho}_{j,r,i}^{(t+1)} &= \overline{\rho}_{j,r,i}^{(t)} - \frac{\eta}{nm} \cdot \sum_{k \in \mathcal{N}(i)} D_k^{-1} \ell_k'^{(t)} \cdot \sigma'(\langle \mathbf{w}_{j,r}^{(t)}, \tilde{\boldsymbol{\xi}}_k \rangle) \cdot \mathbb{1}(y_k = j)\|\boldsymbol{\xi}_i\|_2^2.
\end{aligned}
$$

We define $t_{j,r,i}$ as the final moment $t < T^*$ when $\overline{\rho}_{j,r,i}^{(t)} \leq 0.5\alpha$.

We can express $\overline{\rho}_{j,r,i}^{(\tilde{T})}$ as follows:

$$
\begin{aligned}
\overline{\rho}_{j,r,i}^{(\tilde{T})} &= \overline{\rho}_{j,r,i}^{(t_{j,r,i})} - \underbrace{\frac{\eta}{nm} \cdot \sum_{k \in \mathcal{N}(i)} D_k^{-1} \cdot \ell_k'^{(t_{j,r,i})} \cdot \sigma'(\langle \mathbf{w}_{j,r}^{(t_{j,r,i})}, \tilde{\boldsymbol{\xi}}_k \rangle) \cdot \mathbb{1}(y_k = j)\|\boldsymbol{\xi}_i\|_2^2}_{I_1} \\
&\quad - \underbrace{\sum_{t_{j,r,i} < t < T} \frac{\eta}{nm} \cdot \sum_{k \in \mathcal{N}(i)} D_k^{-1} \cdot \ell_k'^{(t)} \cdot \sigma'(\langle \mathbf{w}_{j,r}^{(t)}, \tilde{\boldsymbol{\xi}}_k \rangle) \cdot \mathbb{1}(y_k = j)\|\boldsymbol{\xi}_i\|_2^2}_{I_2}.
\end{aligned}
\tag{27}
$$

Next, we aim to establish an upper bound for $I_1$:

$$
\begin{aligned}
|I_1| &\leq 2qn^{-1}m^{-1}\eta\left(\max_k \hat{\rho}_{j,r,k}^{(t_{j,r,i})} + 0.5\beta + 8n\sqrt{\frac{\log(4n^2/\delta)}{d}}\alpha\right)^{q-1}\sigma_p^2 d \\
&\leq q2^q n^{-1}m^{-1}\eta\alpha^{q-1}\sigma_p^2 d \leq 0.25\alpha,
\end{aligned}
$$

where we apply Lemmas B.6 and A.1 for the first inequality, utilize the conditions $\beta \leq 0.1\alpha$ and $8n\sqrt{\frac{\log(4n^2/\delta)}{d}}\alpha \leq 0.1\alpha$ for the second inequality, and finally, the constraint $\eta \leq nm/(q2^{q+2}\alpha^{q-2}\sigma_p^2 d)$ for the last inequality.

Second, we bound $I_2$. For $t_{j,r,i} < t < \tilde{T}$ and $y_k = j$, we can lower bound $\langle \mathbf{w}_{j,r}^{(t)}, \tilde{\boldsymbol{\xi}}_k \rangle$ as follows,

$$
\begin{aligned}
\langle \mathbf{w}_{j,r}^{(t)}, \tilde{\boldsymbol{\xi}}_k \rangle &\geq \langle \mathbf{w}_{j,r}^{(0)}, \tilde{\boldsymbol{\xi}}_k \rangle + \hat{\rho}_{j,r,k}^{(t)} - 8n\sqrt{\frac{\log(4n^2/\delta)}{d}}\alpha \\
&\geq -0.5\beta + \frac{1}{4}\frac{p-s}{p+s}\alpha - 8n\sqrt{\frac{\log(4n^2/\delta)}{d}}\alpha \\
&\geq 0.25\alpha,
\end{aligned}
$$

where the first inequality is by Lemma B.6, the second inequality is by $\hat{\rho}_{j,r,i}^{(t)} > \frac{1}{4}\frac{p-s}{p+s}\alpha$ and $\langle \mathbf{w}_{j,r}^{(0)}, \tilde{\boldsymbol{\xi}}_i \rangle \geq -0.5\beta$ due to the definition of $t_{j,r,i}$ and $\beta$, the last inequality is by $\beta \leq 0.1\alpha$ and $8n\sqrt{\frac{\log(4n^2/\delta)}{d}}\alpha \leq 0.1\alpha$. Similarly, for $t_{j,r,i} < t < \tilde{T}$ and $y_k = j$, we can also upper bound $\langle \mathbf{w}_{j,r}^{(t)}, \tilde{\boldsymbol{\xi}}_k \rangle$ as follows,

$$
\begin{aligned}
\langle \mathbf{w}_{j,r}^{(t)}, \tilde{\boldsymbol{\xi}}_k \rangle &\leq \langle \mathbf{w}_{j,r}^{(0)}, \tilde{\boldsymbol{\xi}}_k \rangle + \hat{\rho}_{j,r,k}^{(t)} + 8n\sqrt{\frac{\log(4n^2/\delta)}{d}}\alpha \\
&\leq 0.5\beta + \frac{3}{4}\frac{p-s}{p+s}\alpha + 8n\sqrt{\frac{\log(4n^2/\delta)}{d}}\alpha \\
&\leq 2\alpha,
\end{aligned}
$$

where the first inequality is by Lemma B.6, the second inequality is by induction hypothesis $\hat{\rho}_{j,r,i}^{(t)} \leq \alpha$, the last inequality is by $\beta \leq 0.1\alpha$ and $8n\sqrt{\frac{\log(4n^2/\delta)}{d}}\alpha \leq 0.1\alpha$.

Hence, we can derive the following expression for $I_2$:

$$
\begin{aligned}
|I_2| &\leq \sum_{t_{j,r,i} < t < \tilde{T}} \frac{\eta}{nm} \cdot \sum_{k \in \mathcal{N}(i)} D_k^{-1} \exp(-\sigma(\langle \mathbf{w}_{j,r}^{(t)}, \tilde{\boldsymbol{\xi}}_k \rangle) + 1) \cdot \sigma'(\langle \mathbf{w}_{j,r}^{(t)}, \tilde{\boldsymbol{\xi}}_k \rangle) \cdot \mathbb{1}(y_k = j)\|\boldsymbol{\xi}_i\|_2^2 \\
&\leq \frac{eq2^q\eta T^*}{n} \exp(-\alpha^q/4^q)\alpha^{q-1}\sigma_p^2 d \\
&\leq 0.25T^* \exp(-\alpha^q/4^q)\alpha \\
&\leq 0.25T^* \exp(-\log(T^*)^q)\alpha \\
&\leq 0.25\alpha,
\end{aligned}
$$

where we apply equation 26 for the first inequality, utilize Lemma A.1 for the second, employ the constraint $\eta = O\big(nm/(q2^{q+2}\alpha^{q-2}\sigma_p^2 d)\big)$ in equation 16 for the third, and finally, the conditions $\alpha = 4\log(T^*)$ and $\log(T^*)^q \geq \log(T^*)$ for the subsequent inequalities. By incorporating the bounds of $I_1$ and $I_2$ into equation 27, we conclude the proof for $\overline{\rho}$.

In a similar manner, we can establish that $\gamma_{j,r}^{(\tilde{T})} \leq \alpha$ by using $\eta = O\big(nm/(q2^{q+2}\alpha^{q-2}\|\boldsymbol{\mu}\|_2^2)\big)$ in equation 16. Thus, Proposition B.4 is valid for $t = \tilde{T}$, completing the induction process. As a corollary to Proposition B.4, we identify a crucial characteristic of the loss function during training within the interval $0 \leq t \leq T^*$. This characteristic will play a vital role in the subsequent convergence analysis.

$\square$

## C  TWO STAGE DYNAMICS ANALYSIS

In this section, we employ a two-stage dynamics analysis to investigate the behavior of coefficient iterations. During the first stage, the derivative of the loss function remains almost constant due to the small weight initialization. In the second stage, the derivative of the loss function ceases to be constant, necessitating an analysis that meticulously takes this into account.

## C.1 FIRST STAGE: FEATURE LEARNING VERSUS NOISE MEMORIZATION

**Lemma C.1** (Restatement of Lemma 5.2). *Under the same conditions as Theorem 4.3, in particular if we choose*

$$n \cdot \text{SNR}^q \cdot (n(p+s))^{q/2-1} \geq C \log(6/\sigma_0 \|\boldsymbol{\mu}\|_2) 2^{2q+6} [4 \log(8mn/\delta)]^{(q-1)/2}, \tag{28}$$

*where $C = O(1)$ is a positive constant, there exists time $T_1 = \frac{C \log(6/\sigma_0 \|\boldsymbol{\mu}\|_2) 2^{q+1} m}{\eta \sigma_0^{q-2} \|\boldsymbol{\mu}\|_2^q \Xi^q}$ such that*

- $\max_r \gamma_{j,r}^{(T_1)} \geq 2$ *for* $j \in \{\pm 1\}$.

- $|\rho_{j,r,i}^{(t)}| \leq \sigma_0 \sigma_p \sqrt{d/(n(p+s))}/2$ *for all* $j \in \{\pm 1\}, r \in [m], i \in [n]$ *and* $0 \leq t \leq T_1$.

**Remark C.2.** *In this lemma, we establish that the rate of signal learning significantly outpaces that of noise memorization within GNNs. After a specific number of iterations, the GNN is able to learn the signal from the data at a constant or higher order, while only memorizing a smaller order of noise.*

*Proof of Lemma C.1.* Let us define

$$T_1^+ = \frac{nm\eta^{-1}\sigma_0^{2-q}\sigma_p^{-q}d^{-q/2}(n(p+s))^{(q-2)/2}}{2^{q+4}q[4\log(8mn/\delta)]^{(q-2)/2}}. \tag{29}$$

We will begin by establishing the outcome related to noise memorization. Let $\Psi^{(t)}$ be the maximum value over all $j, r, i$ of $|\rho_{j,r,i}^{(t)}|$, that is, $\Psi^{(t)} = \max_{j,r,i}\{\overline{\rho}_{j,r,i}^{(t)}, -\underline{\rho}_{j,r,i}^{(t)}\}$. We will employ an inductive argument to demonstrate that

$$\Psi^{(t)} \leq \sigma_0 \sigma_p \sqrt{d/(n(p+s))} \tag{30}$$

is valid for the entire range $0 \leq t \leq T_1^+$. By its very definition, it is evident that $\Psi^{(0)} = 0$. Assuming that there exists a value $\tilde{T} \leq T_1^+$ for which equation equation 30 is satisfied for all $0 < t \leq \tilde{T} - 1$, we can proceed as follows.

$$\Psi^{(t+1)} \leq \Psi^{(t)} + \frac{\eta}{nm} \sum_{k \in \mathcal{N}(i)} D_k^{-1} \cdot |\ell_k'^{(t)}| \cdot$$

$$\sigma'\left(\langle \mathbf{w}_{j,r}^{(0)}, \tilde{\boldsymbol{\xi}}_k\rangle + \sum_{i'=1}^n \Psi^{(t)} \cdot \frac{|\langle \boldsymbol{\xi}_{i'}, \tilde{\boldsymbol{\xi}}_k\rangle|}{\|\boldsymbol{\xi}_{i'}\|_2^2} + \sum_{i'=1}^n \Psi^{(t)} \cdot \frac{|\langle \boldsymbol{\xi}_{i'}, \tilde{\boldsymbol{\xi}}_k\rangle|}{\|\boldsymbol{\xi}_{i'}\|_2^2}\right) \cdot \|\boldsymbol{\xi}_i\|_2^2$$

$$\leq \Psi^{(t)} + \frac{\eta}{nm} \cdot \sum_{k \in \mathcal{N}(i)} D_k^{-1} \sigma'\left(\langle \mathbf{w}_{j,r}^{(0)}, \tilde{\boldsymbol{\xi}}_k\rangle + 2 \cdot \sum_{i'=1}^n \Psi^{(t)} \cdot \frac{|\langle \boldsymbol{\xi}_{i'}, \tilde{\boldsymbol{\xi}}_k\rangle|}{\|\boldsymbol{\xi}_{i'}\|_2^2}\right) \cdot \|\boldsymbol{\xi}_i\|_2^2$$

$$= \Psi^{(t)} + \frac{\eta}{nm} \cdot \sum_{k \in \mathcal{N}(i)} D_k^{-1} \cdot$$

$$\sigma'\left(\langle \mathbf{w}_{j,r}^{(0)}, \tilde{\boldsymbol{\xi}}_k\rangle + 2\Psi^{(t)} + 2 \cdot \sum_{i' \neq k}^n \Psi^{(t)} \cdot D_k^{-1} \sum_{k' \in \mathcal{N}(k)} \frac{|\langle \boldsymbol{\xi}_{i'}, \boldsymbol{\xi}_{k'}\rangle|}{\|\boldsymbol{\xi}_{i'}\|_2^2}\right) \cdot \|\boldsymbol{\xi}_i\|_2^2$$

$$\leq \Psi^{(t)} + \frac{\eta q}{nm} \cdot \sum_{k \in \mathcal{N}(i)} D_k^{-1}\left[2 \cdot \sqrt{\log(8mn/\delta)} \cdot \sigma_0 \sigma_p \sqrt{d/(n(p+s))}\right.$$

$$\left. + \left(2 + \frac{4n\sigma_p^2 \cdot \sqrt{d\log(4n^2/\delta)}}{\sigma_p^2 d}\right) \cdot \Psi^{(t)}\right]^{q-1} \cdot 2\sigma_p^2 d$$

$$\leq \Psi^{(t)} + \frac{\eta q}{nm} \cdot \left(2 \cdot \sqrt{\log(8mn/\delta)} \cdot \sigma_0 \sigma_p \sqrt{d/(n(p+s))} + 4\Psi^{(t)}\right)^{q-1} \cdot 2\sigma_p^2 d$$

$$\leq \Psi^{(t)} + \frac{\eta q}{nm} \cdot \left(4 \cdot \sqrt{\log(8mn/\delta)} \cdot \sigma_0 \sigma_p \sqrt{d/(n(p+s))}\right)^{q-1} \cdot 2\sigma_p^2 d,$$

where the second inequality is due to the constraint $|\ell_i'^{(t)}| \leq 1$, the third inequality is derived from Lemmas A.1 and A.7, the fourth inequality is a consequence of the condition $d \geq 16Dn^2 \log(4n^2/\delta)$, and the final inequality is a result of the inductive assumption equation 30. Summing over the sequence $t = 0, 1, \ldots, \tilde{T} - 1$, we obtain

$$
\begin{aligned}
\Psi^{(\tilde{T})} &\leq \tilde{T} \cdot \frac{\eta q}{nm} \cdot \left(4 \cdot \sqrt{\log(8mn/\delta)} \cdot \sigma_0 \sigma_p \sqrt{d/(n(p+s))}\right)^{q-1} \cdot 2\sigma_p^2 d \\
&\leq T_1^+ \cdot \frac{\eta q}{nm} \cdot \left(4 \cdot \sqrt{\log(8mn/\delta)} \cdot \sigma_0 \sigma_p \sqrt{d/(n(p+s))}\right)^{q-1} \cdot 2\sigma_p^2 d \\
&\leq \frac{\sigma_0 \sigma_p \sqrt{d/(n(p+s))}}{2},
\end{aligned}
$$

where the second inequality is justified by $\tilde{T} \leq T_1^+$ in our inductive argument. Hence, by induction, we conclude that $\Psi^{(t)} \leq \sigma_0 \sigma_p \sqrt{d/n(p+s)}/2$ for all $t \leq T_1^+$.

Next, we can assume, without loss of generality, that $j = 1$. Let $T_{1,1}$ represent the final time for $t$ within the interval $[0, T_1^+]$ such that $\max_r \gamma_{1,r}^{(t)} \leq 2$, given $\sigma_0 \leq \sqrt{n(p+s)/d}/\sigma_p$. For $t \leq T_{1,1}$, we have $\max_{j,r,i}\{|\rho_{j,r,i}^{(t)}|\} = O(\sigma_0 \sigma_p \sqrt{d/(n(p+s))}) = O(1)$ and $\max_r \gamma_{1,r}^{(t)} \leq 2$. By applying Lemmas B.8 and B.10, we deduce that $F_{-1}(\mathbf{W}_{-1}^{(t)}, \tilde{\mathbf{x}}_i), F_{+1}(\mathbf{W}_{+1}^{(t)}, \tilde{\mathbf{x}}_i) = O(1)$ for all $i$ with $y_i = 1$. Consequently, there exists a positive constant $C_1$ such that $-\ell_i'^{(t)} \geq C_1$ for all $i$ with $y_i = 1$.

By equation 11, for $t \leq T_{1,1}$ we have

$$
\begin{aligned}
\gamma_{1,r}^{(t+1)} &= \gamma_{1,r}^{(t)} - \frac{\eta}{nm} \cdot \sum_{i=1}^n \ell_i'^{(t)} \cdot \sigma'(\tilde{y}_i \cdot \langle \mathbf{w}_{1,r}^{(0)}, \boldsymbol{\mu} \rangle + \tilde{y}_i \cdot \gamma_{1,r}^{(t)}) \cdot \tilde{y}_i \|\boldsymbol{\mu}\|_2^2 \\
&\geq \gamma_{1,r}^{(t)} + \frac{C_1 \eta}{nm} \cdot \sum_{y_i=1} \sigma'(y_i \Xi \cdot \langle \mathbf{w}_{1,r}^{(0)}, \boldsymbol{\mu} \rangle + y_i \Xi \cdot \gamma_{1,r}^{(t)}) \cdot \frac{p-s}{p+s} \|\boldsymbol{\mu}\|_2^2.
\end{aligned}
$$

Denote $\hat{\gamma}_{1,r}^{(t)} = \gamma_{1,r}^{(t)} + \langle \mathbf{w}_{1,r}^{(0)}, \boldsymbol{\mu} \rangle$ and let $A^{(t)} = \max_r \hat{\gamma}_{1,r}^{(t)}$. Then we have

$$
\begin{aligned}
A^{(t+1)} &\geq A^{(t)} + \frac{C_1 \eta}{nm} \cdot \sum_{y_i=1} \sigma'(\Xi A^{(t)}) \cdot \Xi \|\boldsymbol{\mu}\|_2^2 \\
&\geq A^{(t)} + \frac{C_1 \eta q \|\boldsymbol{\mu}\|_2^2}{4m} \left[\Xi A^{(t)}\right]^{q-1} \Xi \\
&\geq \left(1 + \frac{C_1 \eta q \|\boldsymbol{\mu}\|_2^2}{4m} \left[A^{(0)}\right]^{q-2} \Xi^q\right) A^{(t)} \\
&\geq \left(1 + \frac{C_1 \eta q \sigma_0^{q-2} \|\boldsymbol{\mu}\|_2^q}{2^q m} \Xi^q\right) A^{(t)},
\end{aligned}
$$

where the second inequality arises from the lower bound on the quantity of positive data as established in Lemma A.4, the third inequality is a result of the increasing nature of the sequence $A^{(t)}$, and the final inequality is derived from $A^{(0)} = \max_r \langle \mathbf{w}_{1,r}^{(0)}, \boldsymbol{\mu} \rangle \geq \sigma_0 \|\boldsymbol{\mu}\|_2/2$, as proven in Lemma A.7. Consequently, the sequence $A^{(t)}$ exhibits exponential growth, and we can express it as

$$
\begin{aligned}
A^{(t)} &\geq \left(1 + \frac{C_1 \eta q \sigma_0^{q-2} \|\boldsymbol{\mu}\|_2^q}{2^q m} \Xi^q\right)^t A^{(0)} \\
&\geq \exp\left(\frac{C_1 \eta q \sigma_0^{q-2} \|\boldsymbol{\mu}\|_2^q}{2^{q+1} m} \Xi^q t\right) A^{(0)} \\
&\geq \exp\left(\frac{C_1 \eta q \sigma_0^{q-2} \|\boldsymbol{\mu}\|_2^q}{2^{q+1} m} \Xi^q t\right) \frac{\sigma_0 \|\boldsymbol{\mu}\|_2}{2},
\end{aligned}
$$

where the second inequality is justified by the relation $1 + z \geq \exp(z/2)$ for $z \leq 2$ and our specific conditions on $\eta$ and $\sigma_0$ as listed in Condition 4.1. The last inequality is a consequence of Lemma A.7

and the definition of $A^{(0)}$. Thus, $A^{(t)}$ will attain the value of 2 within $T_1$ iterations, defined as

$$T_1 = \frac{\log(6/\sigma_0\|\boldsymbol{\mu}\|_2)2^{q+1}m}{C_1\eta q\sigma_0^{q-2}\|\boldsymbol{\mu}\|_2^q\Xi^q}.$$

Since $\max_r \gamma_{1,r}^{(t)} \geq A^{(t)} - 1$, $\max_r \gamma_{1,r}^{(t)}$ will reach 2 within $T_1$ iterations. Next, we can confirm that

$$T_1 \leq \frac{nm\eta^{-1}\sigma_0^{2-q}\sigma_p^{-q}d^{-q/2}(n(p+s))^{(q-2)/2}}{2^{q+5}q[4\log(8mn/\delta)]^{(q-1)/2}} = T_1^+/2,$$

where the inequality is consistent with our SNR condition in equation 28. Therefore, by the definition of $T_{1,1}$, we deduce that $T_{1,1} \leq T_1 \leq T_1^+/2$, utilizing the non-decreasing property of $\gamma$. The proof for $j = -1$ follows a similar logic, leading us to the conclusion that $\max_r \gamma_{-1,r}^{(T_{1,-1})} \geq 2$ while $T_{1,-1} \leq T_1 \leq T_1^+/2$, thereby completing the proof.

$\square$

## C.2 SECOND STAGE: CONVERGENCE ANALYSIS

After the first stage and at time step $T_1$ we know that:

$$\mathbf{w}_{j,r}^{(T_1)} = \mathbf{w}_{j,r}^{(0)} + j \cdot \gamma_{j,r}^{(T_1)} \cdot \frac{\boldsymbol{\mu}}{\|\boldsymbol{\mu}\|_2^2} + \sum_{i=1}^n \overline{\rho}_{j,r,i}^{(T_1)} \cdot \frac{\boldsymbol{\xi}_i}{\|\boldsymbol{\xi}_i\|_2^2} + \sum_{i=1}^n \underline{\rho}_{j,r,i}^{(T_1)} \cdot \frac{\boldsymbol{\xi}_i}{\|\boldsymbol{\xi}_i\|_2^2}.$$

And at the beginning of the second stage, we have following property holds:

- $\max_r \gamma_{j,r}^{(T_1)} \geq 2, \forall j \in \{\pm 1\}$.
- $\max_{j,r,i} |\rho_{j,r,i}^{(T_1)}| \leq \hat{\beta}$ where $\hat{\beta} = \sigma_0\sigma_p\sqrt{d/(n(p+s))}/2$.

Lemma 5.1 implies that the learned feature $\gamma_{j,r}^{(t)}$ will not get worse, i.e., for $t \geq T_1$, we have that $\gamma_{j,r}^{(t+1)} \geq \gamma_{j,r}^{(t)}$, and therefore $\max_r \gamma_{j,r}^{(t)} \geq 2$. Now we choose $\mathbf{W}^*$ as follows:

$$\mathbf{w}_{j,r}^* = \mathbf{w}_{j,r}^{(0)} + 2qm\log(2q/\epsilon) \cdot j \cdot \frac{\boldsymbol{\mu}}{\|\boldsymbol{\mu}\|_2^2}.$$

While the context of CNN presents subtle differences from the scenario described in CNN (Cao et al., 2022), we can adapt the same analytical approach to derive the following two lemmas:

**Lemma C.3** ((Cao et al., 2022)). *Under the same conditions as Theorem 4.3, we have that* $\|\mathbf{W}^{(T_1)} - \mathbf{W}^*\|_F \leq \tilde{O}(m^{3/2}\|\boldsymbol{\mu}\|_2^{-1})$.

**Lemma C.4** ((Cao et al., 2022)). *Under the same conditions as Theorem 4.3, we have that*

$$\|\mathbf{W}^{(t)} - \mathbf{W}^*\|_F^2 - \|\mathbf{W}^{(t+1)} - \mathbf{W}^*\|_F^2 \geq (2q-1)\eta L_{\mathcal{S}}(\mathbf{W}^{(t)}) - \eta\epsilon$$

*for all* $T_1 \leq t \leq T^*$.

**Lemma C.5** (Restatement of Lemma 5.3). *Under the same conditions as Theorem 4.3, let* $T = T_1 + \left\lfloor \frac{\|\mathbf{W}^{(T_1)} - \mathbf{W}^*\|_F^2}{2\eta\epsilon} \right\rfloor = T_1 + \tilde{O}(m^3\eta^{-1}\epsilon^{-1}\|\boldsymbol{\mu}\|_2^{-2})$. *Then we have* $\max_{j,r,i} |\rho_{j,r,i}^{(t)}| \leq 2\hat{\beta} = \sigma_0\sigma_p\sqrt{d/(n(p+s))}$ *for all* $T_1 \leq t \leq T$. *Besides,*

$$\frac{1}{t - T_1 + 1}\sum_{s=T_1}^t L_{\mathcal{S}}(\mathbf{W}^{(s)}) \leq \frac{\|\mathbf{W}^{(T_1)} - \mathbf{W}^*\|_F^2}{(2q-1)\eta(t - T_1 + 1)} + \frac{\epsilon}{2q-1}$$

*for all* $T_1 \leq t \leq T$, *and we can find an iterate with training loss smaller than* $\epsilon$ *within* $T$ *iterations.*

*Proof of Lemma C.5.* We adapt the convergence proof for CNN(Cao et al., 2022) to extend the analysis to GNN. By invoking Lemma C.4, for any given time interval $t \in [T_1, T]$, we can deduce that

$$\|\mathbf{W}^{(s)} - \mathbf{W}^*\|_F^2 - \|\mathbf{W}^{(s+1)} - \mathbf{W}^*\|_F^2 \geq (2q-1)\eta L_{\mathcal{S}}(\mathbf{W}^{(s)}) - \eta\epsilon,$$

which is valid for $s \leq t$. Summing over this interval, we arrive at

$$\sum_{s=T_1}^{t} L_{\mathcal{S}}(\mathbf{W}^{(s)}) \leq \frac{\|\mathbf{W}^{(T_1)} - \mathbf{W}^*\|_F^2 + \eta\epsilon(t - T_1 + 1)}{(2q-1)\eta}. \tag{31}$$

This inequality holds for all $T_1 \leq t \leq T$. Dividing both sides of equation 31 by $(t - T_1 + 1)$, we obtain

$$\frac{1}{t - T_1 + 1} \sum_{s=T_1}^{t} L_{\mathcal{S}}(\mathbf{W}^{(s)}) \leq \frac{\|\mathbf{W}^{(T_1)} - \mathbf{W}^*\|_F^2}{(2q-1)\eta(t - T_1 + 1)} + \frac{\epsilon}{2q-1}.$$

By setting $t = T$, we find that

$$\frac{1}{T - T_1 + 1} \sum_{s=T_1}^{T} L_{\mathcal{S}}(\mathbf{W}^{(s)}) \leq \frac{\|\mathbf{W}^{(T_1)} - \mathbf{W}^*\|_F^2}{(2q-1)\eta(T - T_1 + 1)} + \frac{\epsilon}{2q-1} \leq \frac{3\epsilon}{2q-1} < \epsilon,$$

where we utilize the condition that $q > 2$ and the specific choice of $T = T_1 + \left\lfloor \frac{\|\mathbf{W}^{(T_1)} - \mathbf{W}^*\|_F^2}{2\eta\epsilon} \right\rfloor$. Since the mean value is less than $\epsilon$, it follows that there must exist a time interval $T_1 \leq t \leq T$ for which $L_{\mathcal{S}}(\mathbf{W}^{(t)}) < \epsilon$.

Finally, we aim to demonstrate that $\max_{j,r,i} |\rho_{j,r,i}^{(t)}| \leq 2\hat{\beta}$ holds for all $t \in [T_1, T]$. By inserting $T = T_1 + \left\lfloor \frac{\|\mathbf{W}^{(T_1)} - \mathbf{W}^*\|_F^2}{2\eta\epsilon} \right\rfloor$ into equation equation 31, we obtain

$$\sum_{s=T_1}^{T} L_{\mathcal{S}}(\mathbf{W}^{(s)}) \leq \frac{2\|\mathbf{W}^{(T_1)} - \mathbf{W}^*\|_F^2}{(2q-1)\eta} = \tilde{O}(\eta^{-1}m^3\|\boldsymbol{\mu}\|_2^2), \tag{32}$$

where the inequality is a consequence of $\|\mathbf{W}^{(T_1)} - \mathbf{W}^*\|_F \leq \tilde{O}(m^{3/2}\|\boldsymbol{\mu}\|_2^{-1})$ as shown in Lemma C.3.

Let's define $\Psi^{(t)} = \max_{j,r,i} |\rho_{j,r,i}^{(t)}|$. We will employ induction to prove $\Psi^{(t)} \leq 2\hat{\beta}$ for all $t \in [T_1, T]$. At $t = T_1$, by the definition of $\hat{\beta}$, it is clear that $\Psi^{(T_1)} \leq \hat{\beta} \leq 2\hat{\beta}$.

Assuming that there exists $\tilde{T} \in [T_1, T]$ such that $\Psi^{(t)} \leq 2\hat{\beta}$ for all $t \in [T_1, \tilde{T} - 1]$, we can consider $t \in [T_1, \tilde{T} - 1]$. Using the expression:

$$\rho_{j,r,i}^{(t+1)} = \rho_{j,r,i}^{(t)} - \frac{\eta}{nm} \cdot \sum_{k \in \mathcal{N}(i)} D_k^{-1} \ell_k'^{(t)}$$

$$\sigma'\left( \langle \mathbf{w}_{j,r}^{(0)}, \tilde{\boldsymbol{\xi}}_k \rangle + \sum_{i'=1}^{n} \overline{\rho}_{j,r,i'}^{(t)} \frac{\langle \boldsymbol{\xi}_{i'}, \tilde{\boldsymbol{\xi}}_k \rangle}{\|\boldsymbol{\xi}_{i'}\|_2^2} + \sum_{i'=1}^{n} \underline{\rho}_{j,r,i'}^{(t)} \frac{\langle \boldsymbol{\xi}_{i'}, \tilde{\boldsymbol{\xi}}_k \rangle}{\|\boldsymbol{\xi}_{i'}\|_2^2} \right) \cdot \|\boldsymbol{\xi}_i\|_2^2 \tag{33}$$

we can proceed to analyze:

$$\Psi^{(t+1)} \leq \Psi^{(t)} + \max_{j,r,i} \left\{ \frac{\eta}{nm} \cdot \sum_{k \in \mathcal{N}(i)} D_k^{-1} |\ell_k'^{(t)}| \cdot \sigma' \left( \langle \mathbf{w}_{j,r}^{(0)}, \tilde{\boldsymbol{\xi}}_k \rangle + 2 \sum_{i'=1}^{n} \Psi^{(t)} \cdot \frac{|\langle \boldsymbol{\xi}_{i'}, \tilde{\boldsymbol{\xi}}_k \rangle|}{\|\boldsymbol{\xi}_{i'}\|_2^2} \right) \cdot \|\boldsymbol{\xi}_i\|_2^2 \right\}$$

$$= \Psi^{(t)} + \max_{j,r,i} \left\{ \frac{\eta}{nm} \cdot \sum_{k \in \mathcal{N}(i)} D_k^{-1} |\ell_k'^{(t)}| \cdot \right.$$

$$\left. \sigma' \left( \langle \mathbf{w}_{j,r}^{(0)}, \tilde{\boldsymbol{\xi}}_k \rangle + 2\Psi^{(t)} + 2 \sum_{i' \neq k'}^{n} \Psi^{(t)} \cdot D_k^{-1} \sum_{k' \in \mathcal{N}(k)} \frac{|\langle \boldsymbol{\xi}_{i'}, \boldsymbol{\xi}_{k'} \rangle|}{\|\boldsymbol{\xi}_{i'}\|_2^2} \right) \cdot \|\boldsymbol{\xi}_i\|_2^2 \right\}$$

$$\leq \Psi^{(t)} + \frac{\eta q}{nm} \cdot \max_i \sum_{k \in \mathcal{N}(i)} D_k^{-1} |\ell_k'^{(t)}| \cdot \left[ 2 \cdot \sqrt{\log(8mn/\delta)} \cdot \sigma_0 \sigma_p \sqrt{d/(n(p+s))} \right.$$

$$\left. + \left( 2 + \frac{4n\sigma_p^2 \cdot \sqrt{d \log(4n^2/\delta)}}{\sigma_p^2 d/2} \right) \cdot \Psi^{(t)} \right]^{q-1} \cdot 2\sigma_p^2 d$$

$$\leq \Psi^{(t)} + \frac{\eta q}{nm} \cdot \max_i \sum_{k \in \mathcal{N}(i)} D_k^{-1} |\ell_k'^{(t)}| \cdot$$

$$\left( 2 \cdot \sqrt{\log(8mn/\delta)} \cdot \sigma_0 \sigma_p \sqrt{d/(n(p+s))} + 4 \cdot \Psi^{(t)} \right)^{q-1} \cdot 2\sigma_p^2 d.$$

The second inequality is derived from Lemmas A.1 and A.7, while the final inequality is based on the assumption that $d \geq 16n^2 \log(4n^2/\delta)$. By taking a telescoping sum, we can express the following:

$$\Psi^{(T)} \overset{(i)}{\leq} \Psi^{(T_1)} + \frac{\eta q}{nm} \sum_{s=T_1}^{\tilde{T}-1} \max_i \sum_{k \in \mathcal{N}(i)} D_k^{-1} |\ell_k'^{(t)}| \tilde{O}(\sigma_p^2 d) \hat{\beta}^{q-1}$$

$$\overset{(ii)}{\leq} \Psi^{(T_1)} + \frac{\eta q}{nm} \tilde{O}(\sigma_p^2 d) \hat{\beta}^{q-1} \sum_{s=T_1}^{\tilde{T}-1} \max_i \sum_{k \in \mathcal{N}(i)} D_k^{-1} \ell_k^{(s)}$$

$$\overset{(iii)}{\leq} \Psi^{(T_1)} + \tilde{O}(\eta m^{-1} \sigma_p^2 d) \hat{\beta}^{q-1} \sum_{s=T_1}^{\tilde{T}-1} L_{\mathcal{S}}(\mathbf{W}^{(s)})$$

$$\overset{(iv)}{\leq} \Psi^{(T_1)} + \tilde{O}(m^2 \text{SNR}^{-2}) \hat{\beta}^{q-1}$$

$$\overset{(v)}{\leq} \hat{\beta} + \tilde{O}(m^2 n^{2/q} (n(p+s))^{1-2/q} \hat{\beta}^{q-2}) \hat{\beta}$$

$$\overset{(vi)}{\leq} 2\hat{\beta},$$

where (i) follows from our induction assumption that $\Psi^{(t)} \leq 2\hat{\beta}$, (ii) is derived from the relationship $|\ell'| \leq \ell$, (iii) is obtained by the sum of $\max_i \sum_{k \in \mathcal{N}(i)} D_k^{-1} \leq \sum_i \ell_i^{(s)} = nL_{\mathcal{S}}(\mathbf{W}^{(s)})$, (iv) is due to the summation of $\sum_{s=T_1}^{\tilde{T}-1} L_S(\mathbf{W}^{(s)}) \leq \sum_{s=T_1}^{T} L_S(\mathbf{W}^{(s)}) = \tilde{O}(\eta^{-1} m^3 \|\boldsymbol{\mu}\|_2^2)$ as shown in equation 32, (v) is based on the condition $n\text{SNR}^q \cdot (n(p+s))^{q/2-1} \geq \tilde{\Omega}(1)$, and (vi) follows from the definition of $\hat{\beta} = \sigma_0 \sigma_p \sqrt{d/(n(p+s))}/2$ and $\tilde{O}(m^2 n^{2/q} (n(p+s))^{1-2/q} \hat{\beta}^{q-2}) = \tilde{O}(m^2 n^{2/q} (n(p+s))^{1-2/q} (\sigma_0 \sigma_p \sqrt{d/(n(p+s))})^{q-2}) \leq 1$.

Thus, we conclude that $\Psi^{(\tilde{T})} \leq 2\hat{\beta}$, completing the induction and establishing the desired result. $\qquad \square$

### C.3 POPULATION LOSS

Consider a new data point $(\mathbf{x}, y)$ drawn from the SNM-SBM distribution. Without loss of generality, we suppose that the first patch is the signal patch and the second patch is the noise patch, i.e., $\mathbf{x} =$

$[y \cdot \boldsymbol{\mu}, \boldsymbol{\xi}]$. Moreover, by the signal-noise decomposition, the learned neural network has parameter:

$$\mathbf{w}_{j,r}^{(t)} = \mathbf{w}_{j,r}^{(0)} + j \cdot \gamma_{j,r}^{(t)} \cdot \frac{\boldsymbol{\mu}}{\|\boldsymbol{\mu}\|_2^2} + \sum_{i=1}^{n} \overline{\rho}_{j,r,i}^{(t)} \cdot \frac{\boldsymbol{\xi}_i}{\|\boldsymbol{\xi}_i\|_2^2} + \sum_{i=1}^{n} \underline{\rho}_{j,r,i}^{(t)} \cdot \frac{\boldsymbol{\xi}_i}{\|\boldsymbol{\xi}_i\|_2^2}$$

for $j \in \{\pm 1\}$ and $r \in [m]$.

Although the framework of CNN diverges in certain nuances from the situation of CNN outlined in (Cao et al., 2022), we are able to employ a similar analytical methodology to deduce the subsequent two lemmas:

**Lemma C.6.** *Under the same conditions as Theorem 4.3, we have that* $\max_{j,r} |\langle \mathbf{w}_{j,r}^{(t)}, \tilde{\boldsymbol{\xi}}_i \rangle| \leq 1/2$ *for all* $0 \leq t \leq T$, *and* $i \in [n]$.

**Lemma C.7.** *Under the same conditions as Theorem 4.3, with probability at least* $1 - 4mT \cdot \exp(-C_2^{-1}\sigma_0^{-2}\sigma_p^{-2}d^{-1}n(p+s))$, *we have that* $\max_{j,r} |\langle \mathbf{w}_{j,r}^{(t)}, \tilde{\boldsymbol{\xi}} \rangle| \leq 1/2$ *for all* $0 \leq t \leq T$, *where* $C_2 = \tilde{O}(1)$.

**Lemma C.8** (Restatement of Lemma 5.4). *Let $T$ be defined in Lemma 5.2 respectively. Under the same conditions as Theorem 4.3, for any $0 \leq t \leq T$ with $L_S(\mathbf{W}^{(t)}) \leq 1$, it holds that $L_{\mathcal{D}}(\mathbf{W}^{(t)}) \leq c_1 \cdot L_S(\mathbf{W}^{(t)}) + \exp(-c_2 n^2)$.*

*Proof of Lemma C.8.* Consider the occurrence of event $\mathcal{E}$, defined as the condition under which Lemma C.7 is satisfied. We can then express the loss $L_{\mathcal{D}}(\mathbf{W}^{(t)})$ as a sum of two components:

$$\mathbb{E}\big[\ell\big(yf(\mathbf{W}^{(t)}, \tilde{\mathbf{x}})\big)\big] = \underbrace{\mathbb{E}[\mathbb{1}(\mathcal{E})\ell\big(yf(\mathbf{W}^{(t)}, \tilde{\mathbf{x}})\big)]}_{\text{Term } I_1} + \underbrace{\mathbb{E}[\mathbb{1}(\mathcal{E}^c)\ell\big(yf(\mathbf{W}^{(t)}, \tilde{\mathbf{x}})\big)]}_{\text{Term } I_2}. \tag{34}$$

Next, we proceed to establish bounds for $I_1$ and $I_2$.

**Bounding $I_1$:** Given that $L_S(\mathbf{W}^{(t)}) \leq 1$, there must be an instance $(\tilde{\mathbf{x}}_i, y_i)$ for which $\ell\big(y_i f(\mathbf{W}^{(t)}, \tilde{\mathbf{x}}_i)\big) \leq L_S(\mathbf{W}^{(t)}) \leq 1$, leading to $y_i f(\mathbf{W}^{(t)}, \tilde{\mathbf{x}}_i) \geq 0$. Hence, we obtain:

$$\exp(-y_i f(\mathbf{W}^{(t)}, \tilde{\mathbf{x}}_i)) \overset{(i)}{\leq} 2\log\big(1 + \exp(-y_i f(\mathbf{W}^{(t)}, \tilde{\mathbf{x}}_i))\big) = 2\ell\big(y_i f(\mathbf{W}^{(t)}, \tilde{\mathbf{x}}_i)\big) \leq 2L_S(\mathbf{W}^{(t)}), \tag{35}$$

where (i) follows from the inequality $z \leq 2\log(1+z), \forall z \leq 1$. If event $\mathcal{E}$ occurs, we deduce:

$$|yf(\mathbf{W}^{(t)}, \tilde{\mathbf{x}}^{(2)}) - y_i f(\mathbf{W}^{(t)}, \tilde{\mathbf{x}}_i^{(2)})| \leq \frac{1}{m}\sum_{j,r}\sigma(\langle \mathbf{w}_{j,r}, \tilde{\boldsymbol{\xi}}_i \rangle) + \frac{1}{m}\sum_{j,r}\sigma(\langle \mathbf{w}_{j,r}, \tilde{\boldsymbol{\xi}} \rangle)$$
$$\leq 1. \tag{36}$$

Here, $f(\mathbf{W}^{(t)}, \tilde{\mathbf{x}}^{(2)})$ refers to the input $\tilde{\mathbf{x}} = [0, \tilde{\mathbf{x}}^{(2)}]$. The second inequality is justified by Lemmas C.7 and C.6. Consequently, we have:

$$I_1 \leq \mathbb{E}[\mathbb{1}(\mathcal{E})\exp(-yf(\mathbf{W}^{(t)}, \tilde{\mathbf{x}}))]$$
$$= \mathbb{E}[\mathbb{1}(\mathcal{E})\exp(-y_i f(\mathbf{W}^{(t)}, \tilde{\mathbf{x}}^{(1)}))\exp(-y_i f(\mathbf{W}^{(t)}, \tilde{\mathbf{x}}^{(2)}))]$$
$$\leq 2e \cdot C \cdot \mathbb{E}[\mathbb{1}(\mathcal{E})\exp(-y_i f(\mathbf{W}^{(t)}, \tilde{\mathbf{x}}_i^{(1)}))\exp(-y_i f(\mathbf{W}^{(t)}, \tilde{\mathbf{x}}_i^{(2)}))]$$
$$\leq 2e \cdot \mathbb{E}[\mathbb{1}(\mathcal{E})L_S(\mathbf{W}^{(t)})],$$

where the inequalities follow from the properties of cross-entropy loss, equation 36, Lemma A.4, and equation 35. The constant $c_1$ encapsulates the factors in the derivation.

**Estimating $I_2$:** We now turn our attention to the second term $I_2$. By selecting an arbitrary training data point $(\mathbf{x}_{i'}, y_{i'})$ with $y_{i'} = y$, we can derive the following:

$$
\begin{aligned}
\ell\big(yf(\mathbf{W}^{(t)}, \tilde{\mathbf{x}})\big) &\leq \log(1 + \exp(F_{-y}(\mathbf{W}^{(t)}, \tilde{\mathbf{x}}))) \\
&\leq 1 + F_{-y}(\mathbf{W}^{(t)}, \tilde{\mathbf{x}}) \\
&= 1 + \frac{1}{m} \sum_{j=-y, r \in [m]} \sigma(\langle \mathbf{w}_{j,r}^{(t)}, \tilde{y}\boldsymbol{\mu} \rangle) + \frac{1}{m} \sum_{j=-y, r \in [m]} \sigma(\langle \mathbf{w}_{j,r}^{(t)}, \tilde{\boldsymbol{\xi}} \rangle) \\
&\leq 1 + F_{-y_i}(\mathbf{W}_{-y_{i'}}, \tilde{\mathbf{x}}_{i'}) + \frac{1}{m} \sum_{j=-y, r \in [m]} \sigma(\langle \mathbf{w}_{j,r}^{(t)}, \tilde{\boldsymbol{\xi}} \rangle) \\
&\leq 2 + \frac{1}{m} \sum_{j=-y, r \in [m]} \sigma(\langle \mathbf{w}_{j,r}^{(t)}, \tilde{\boldsymbol{\xi}} \rangle) \\
&\leq 2 + \tilde{O}((\sigma_0 \sqrt{d})^q) \|\tilde{\boldsymbol{\xi}}\|^q, \tag{37}
\end{aligned}
$$

where the inequalities follow from the properties of the cross-entropy loss and the constraints defined in Lemma B.8. The last inequality is a result of the boundedness of the inner product with $\tilde{\boldsymbol{\xi}}$. Continuing, we have:

$$
\begin{aligned}
I_2 &\leq \sqrt{\mathbb{E}[\mathbb{1}(\mathcal{E}^c)]} \cdot \sqrt{\mathbb{E}\left[\ell\big(yf(\mathbf{W}^{(t)}, \tilde{\mathbf{x}})\big)^2\right]} \\
&\leq \sqrt{\mathbb{P}(\mathcal{E}^c)} \cdot \sqrt{4 + \tilde{O}((\sigma_0 \sqrt{d})^{2q}) \mathbb{E}[\|\tilde{\boldsymbol{\xi}}\|_2^{2q}]} \\
&\leq \exp\left[-\tilde{\Omega}\left(\frac{\sigma_0^{-2} \sigma_p^{-2}}{d^{-1} n(p+s)}\right) + \text{polylog}(d)\right] \\
&\leq \exp(-c_1 n^2),
\end{aligned}
$$

where $c_1$ is a constant, the first inequality is by Cauchy-Schwartz inequality, the second inequality is by equation 37, the third inequality is by Lemma C.7 and the fact that $\sqrt{4 + \tilde{O}((\sigma_0 \sqrt{d})^{2q}) \mathbb{E}[\|\tilde{\boldsymbol{\xi}}\|_2^{2q}]} = O(\text{poly}(d))$, and the last inequality is by our condition $\sigma_0 \leq \tilde{O}(m^{-2/(q-2)} n^{-1}) \cdot (\sigma_p \sqrt{d/(n(p+s))})^{-1}$ in Condition 4.1. Plugging the bounds of $I_1$, $I_2$ completes the proof. $\qquad\square$

# D    ADDITIONAL EXPERIMENTAL PROCEDURES AND RESULTS

## D.1    DATASET IN NODE CLASSIFICATION

In Figure 1, we execute node classification experiments on three frequently used citation networks: Cora, Citeseer, and Pubmed Kipf & Welling (2016a). Detailed information about these datasets is provided below and summarized in Table 1.

Table 1: Details of Datasets

| Dataset | Nodes | Edges | Classes | Features | Train/Val/Test |
|---------|-------|-------|---------|----------|----------------|
| Cora | 2,708 | 5,429 | 7 | 1,433 | 0.05/0.18/0.37 |
| Citeseer | 3,327 | 4,732 | 6 | 3,703 | 0.04/0.15/0.30 |
| Pubmed | 19,717 | 44,338 | 3 | 500 | 0.003/0.03/0.05 |

- The Cora dataset includes 2,708 scientific publications, each categorized into one of seven classes, connected by 5,429 links. Each publication is represented by a binary word vector, which denotes the presence or absence of a corresponding word from a dictionary of 1,433 unique words.

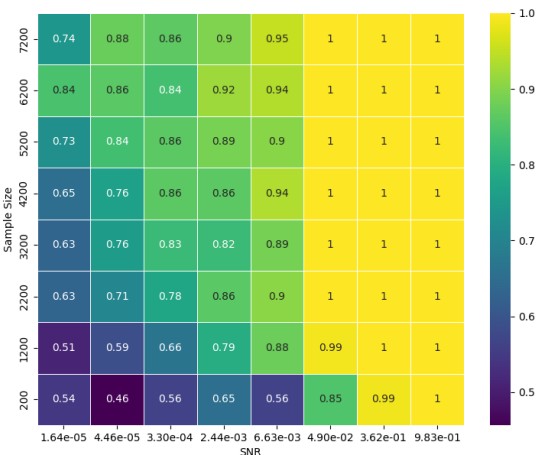

Figure 7: Test accuracy heatmap for GCNs after training.

- The Citeseer dataset comprises 3,312 scientific publications, each classified into one of six classes, connected by 4,732 links. Each publication is represented by a binary word vector, indicating the presence or absence of a corresponding word from a dictionary that includes 3,703 unique words.

- The Pubmed Diabetes dataset includes 19,717 scientific publications related to diabetes, drawn from the PubMed database and classified into one of three classes. The citation network is made up of 44,338 links. Each publication is represented by a TF-IDF weighted word vector from a dictionary consisting of 500 unique words.

### D.2 PARALLELS BETWEEN OUR DATA MODEL AND REAL-WORLD DATASET

These datasets (Cora, Citeseer, and Pubmed) employ a bag-of-words feature representation, typically represented by one-hot vectors, thereby ensuring orthogonality between features. We can conceptually divide words into two categories: label-relevant and label-irrelevant. For example, words like "algorithm" or "neural network" are label-relevant to the subject of computer science, while general words like "study" or "approach" are label-irrelevant. In our SNM, $\mu$ represents label-relevant features, while $\xi$ represents label-irrelevant ones.

Furthermore, the datasets Wiki-CS, Amazon-Computers, Amazon-Photo, Coauthor-CS, and Coauthor-Physics Shchur et al. (2018) also parallels with our theoretical model and we provide the more discussion as follows:

- Coauthor CS (Computer Science) & Coauthor Physics (Coauthor Phy.): The dataset typically includes features based on the keywords of an author's papers, and the task is often to predict each author's research field or interests based on their publication record and collaboration network.

- Amazon Computers & Amazon Photo: Node features are derived from product reviews, and the classification task involves predicting product categories based on the co-purchase relationships and review data.

- WikiCS Node features could be derived from the text of the articles, such as word vectors. The classification task usually involves categorizing articles into different areas or subjects within Computer Science based on their content and the article network structure.

We have broadened our analysis to include the measurement of cosine similarity between two equal-sized parts of node features (excluding the final feature for odd-sized representations) across a diverse range of datasets. This extended analysis bolsters the orthogonality relation posited in our model. The results are presented in Table 2.

| Dataset | Feature Dimension | Cossin Similarity |
|---|---|---|
| Cora | 1433 | $1.57 \times 10^{-5}$ |
| Citeseer | 3703 | $3.99 \times 10^{-6}$ |
| Pubmed | 500 | $2.00 \times 10^{-4}$ |
| Coauthor CS | 6805 | $2.28 \times 10^{-6}$ |
| Coauthor Phy. | 8451 | $1.08 \times 10^{-6}$ |
| Amazon Comp. | 767 | $9.00 \times 10^{-4}$ |
| Amazon Photo | 745 | $9.00 \times 10^{-4}$ |
| WikiCS | 300 | $1.00 \times 10^{-4}$ |

Table 2: Cosine similarity analysis of node features across various datasets.

### D.3 Phase transition in GCN

In Figure 5, we illustrated the variance in test accuracy between CNN and GCN within a chosen range of SNR and sample numbers, where GCN was shown to achieve near-perfect test accuracy. Here, we broaden the SNR range towards the smaller end and display the corresponding phase diagram of GCN in Figure 7. When the SNR is exceedingly small, we observe that GCNs return lower test accuracy, suggesting the possibility of a phase transition in the test accuracy of GCNs.

### D.4 Software and hardware

We implement our methods with PyTorch. For the software and hardware configurations, we ensure the consistent environments for each datasets. We run all the experiments on Linux servers with NVIDIA V100 graphics cards with CUDA 11.2.

