# OpenReview forum: "Graph Neural Networks Provably Benefit from Structural Information: A Feature Learning Perspective"
_ICLR.cc/2024/Conference — Submitted to ICLR 2024_

### Official Review · Reviewer_Q462 · 2023-10-24

**Soundness:** 3 good
**Presentation:** 2 fair
**Contribution:** 2 fair
**Rating:** 3
**Confidence:** 3

**Summary:**

This paper studies why GNNs work from the perspective of feature learning theory, which is proposed by recent work to analyze why CNNs work. This paper identifies the convergence conditions for GNNs under the defined specific data generation model, i.e., SNM-SBM. Overall, this work justifies why GNNs can achieve better generalization ability than CNNs on graph data from the perspective of feature learning theory.

**Strengths:**

1. The motivation for analyzing why GNNs work theoretically is important and potentially inspiring for better GNN designs.

2. The formulation of the feature learning theory in GNNs is grounded and well-presented.

3. The simple simulation experiments are clearly shown.

**Weaknesses:**

1. (I am not an expert in deep learning theory, so my assessment of this might be less confident.) The novelty of the proposed method seems to be limited. According to my preliminary check, the main theory and proof of this work follow the prior feature learning theory paper closely [1]. It seems to be directly adapted from [1]. Could the authors summarize the main difference/contributions of this submission, compared to [1]?

2. The main results of why GNNs work are not surprising and not adding new insights to the community. It is well-known that GNNs can help smooth the node features corresponding to the same class, thus making the classification tasks easier [2]. In this sense, the results from this paper do not add new insights or provide potential future topics in the field.

3. The experimental results are also widely observed by previous work in the community.

[1] Cao, Yuan, et al. "Benign overfitting in two-layer convolutional neural networks." Advances in neural information processing systems 35 (2022): 25237-25250.

[2] Li, Qimai, Zhichao Han, and Xiao-Ming Wu. "Deeper insights into graph convolutional networks for semi-supervised learning." Proceedings of the AAAI conference on artificial intelligence. Vol. 32. No. 1. 2018.

**Questions:**

N/A

Typo: "by" above Eq. (3)

---

> ### Author Response · Authors · 2023-11-18
> **Response to Reviewer Q462 [Part I]**
>
> Thank you for your thoughtful review and feedback on our submission. We appreciate your recognition of the importance and motivation of our work in analyzing GNNs from a feature learning theory perspective.
>
> We would like to first clarify that this paper focuses on the theoretical understanding of GCNs. Our goal is not to propose novel methods or introduce groundbreaking insights, but to **rigorously back up the understanding of GCN through theoretical analysis**. We provide detailed responses to your comments on the weaknesses below.
>
> > **Weakness 1: (I am not an expert in deep learning theory, so my assessment of this might be less confident.) The novelty of the proposed method seems to be limited. According to my preliminary check, the main theory and proof of this work follow the prior feature learning theory paper closely [4]. It seems to be directly adapted from [4]. Could the authors summarize the main difference/contributions of this submission, compared to [4]?**
>
> Our work is motivated by recent works [1,2], which considered data generated from a stochastic block model (same data model as studied in our work), and studied (non-)linear separability of the data points after graph convolutions. However, these works did not study the training of GCNs, therefore cannot demonstrate that a GCN can really be trained to separate the data (this is non-trivial because of the non-convexity of the optimization problem). Moreover, it is also not studied yet whether the trained GCN can generalize well on test data. This is the major motivation of our work.
>
> In a separate line of works [3,4,5], a theoretical framework to analyze the “benign overfitting” phenomenon has been proposed. However, these works have a completely different focus, and their results are not directly related to GCNs or stochastic block models. We believe one of the contributions of our work is that we identify that the analysis framework of the “benign overfitting” phenomenon can be utilized in studying training GCNs on data given by stochastic block models.
>
> Moreover, we would also like to clarify that although we are inspired by the theoretical framework in [3,4,5], our analysis is different from [3,4,5] due to a significantly different problem setting. Our main contributions compared to [4] include:
>
> - **Different iterative coefficient dynamics**: In our study, graph convolution aggregates information from neighboring nodes to the central node. This process often leads to a loss of statistical concentration for the aggregated noise vectors and labels. To overcome this challenge, we introduce a density threshold by $p,s = \Omega( \sqrt{\log(n)/n})$. This condition helps stabilize the feature learning dynamics.
>
> - **Emphasizing the role of graph convolution in diverging learning speeds between signal learning and noise memorization**: We introduce “homophily” by setting $p > s$, which helps convolution process effectively integrates neighborhood information and provides technical support for graph convolution to effectively utilize structural information.
>
> - **Addressing the challenge in generalization analysis of graph neural networks**. We confront the difficulty of analyzing edge connections among the test dataset in GNNs. To do this, we introduce an expectation over the distribution for a single data point. Specifically, we consider a new data point  (x, y) drawn from the distribution SNM-SBM. Then we calculate the test error but taking expectation over the new data point.

---

> ### Author Response · Authors · 2023-11-18
> **Response to Reviewer Q462 [Part II]**
>
> > **Weakness 2: The main results of why GNNs work are not surprising and not adding new insights to the community. It is well-known that GNNs can help smooth the node features corresponding to the same class, thus making the classification tasks easier [6]. In this sense, the results from this paper do not add new insights or provide potential future topics in the field.**
>
> We appreciate the opportunity to compare our work with the findings in [6]. It is important to emphasize that the results in [6] do not directly lead to our conclusions, especially from a theoretical standpoint. The differences are substantial in several key aspects:
>
> 1. **Study of Training Dynamics in GCN**:
>
> - Our Work: We provide a detailed analysis of the training dynamics of GCNs. Starting from the weight initialization, we track the evolution of weights throughout the training process. This approach allows us to understand not just the capacity of GCNs to fit training data, but the actual process by which gradient descent optimizes these parameters, a critical aspect in understanding the learning behavior of GCNs.
>
> - Reference [6]: The work did not delve into the training dynamics of GCN. While it acknowledges certain parameters that enable GCNs to fit training data, it does not investigate whether gradient descent can successfully identify and optimize these parameters during training.
>
> 2. **Generalization of GCN**:
>
> - Our Work: Beyond demonstrating the ability of GCNs to fit training data, we extensively analyze their generalization to test datasets. We not only establish conditions under which GCNs can generalize effectively but also quantify the extent to which GCNs can outperform CNNs in terms of generalization.
>
> - Reference [6]: Their study does not address the generalization of GCNs. Fitting to training data does not necessarily imply effective generalization to unseen test data, an aspect we thoroughly investigate in our research.
>
> ___
>
> > **Weakness 3: The experimental results are also widely observed by previous work in the community.**
>
> We appreciate the opportunity to clarify the context and purpose of our experimental simulations. It's important to understand that our primary objective with these experiments is not to uncover new experimental findings, but rather to validate and demonstrate the practical applicability of our theoretical results.
>
> - **Inspiration for Theoretical Study**: Figure 1 in our paper serves as an inspirational basis for our theoretical study. It is designed to provide an initial visualization and understanding of the concepts that we later delve into from a theoretical standpoint.
>
> - **Verification of Theoretical Results**: The experimental results presented in Figures 3, 4, and 5 are specifically aimed at verifying our theoretical findings. These experiments are carefully designed to align with the theoretical framework we propose.
>
> ___
>
> > **Typo: "by" above Eq. (3)**
>
> We thank you for pointing out the typo above Eq. (3). We have modified it.
>
> ___
>
>
> **References**
>
> [1] Baranwal, Aseem, Kimon Fountoulakis, and Aukosh Jagannath. "Graph Convolution for Semi-Supervised Classification: Improved Linear Separability and Out-of-Distribution Generalization." International Conference on Machine Learning. PMLR, 2021.
>
> [2] Baranwal, Aseem, Kimon Fountoulakis, and Aukosh Jagannath. "Effects of Graph Convolutions in Multi-layer Networks." The Eleventh International Conference on Learning Representations. 2022.
>
> [3] Frei, Spencer, Niladri S. Chatterji, and Peter Bartlett. "Benign overfitting without linearity: Neural network classifiers trained by gradient descent for noisy linear data." Conference on Learning Theory. PMLR, 2022.
>
> [4] Cao, Yuan, et al. "Benign overfitting in two-layer convolutional neural networks." Advances in neural information processing systems 35 (2022): 25237-25250.
>
> [5]  Kou, Yiwen, Zixiang Chen, Yuanzhou Chen, and Quanquan Gu. "Benign Overfitting in Two-layer ReLU Convolutional Neural Networks." International Conference on Machine Learning (2023).
>
> [6] Li, Qimai, Zhichao Han, and Xiao-Ming Wu. "Deeper insights into graph convolutional networks for semi-supervised learning." Proceedings of the AAAI conference on artificial intelligence. Vol. 32. No. 1. 2018.

---

> ### Comment · Reviewer_Q462 · 2023-11-22
>
> Dear authors,
>
> Thank you for your detailed responses to my raised questions. The difference of this work to prior feature learning theory papers is more clear to me. However, the theoretical results and the insights can be conveyed to the graph learning community are still not novel and informative. Thanks.

---

> > ### Author Response · Authors · 2023-11-22
> >
> > Thank you for your feedback. We are pleased to see that the distinction of our work from prior feature learning theory papers is now clearer.
> >
> > Regarding the significance of our theoretical results and insights, we would like to emphasize our appreciation for the progress made by previous works studying the role of graph convolution [1,2,3]. Our paper makes significant progress in two aspects not covered by [1,2,3]:
> >
> > 1. Our research demonstrates how graph convolution achieves better learning considering full gradient descent, overcoming the non-convex optimization challenge. This is a non-trivial contribution to the graph learning community as it delves into the training dynamics of gradient descent. This goes a step further than previous works [1,2,3], which did not consider the entire gradient descent process.
> >
> > 2. We can provably claim that, under the same conditions, the test error of GNNs after training is much smaller than that of CNNs. This result is important for the graph learning community as it highlights a clear gap between the test errors of GNNs and CNNs. Demonstrating this gap rigorously in generalization is one of the best and clear ways to show the superiority of GNNs over CNNs, with the help by graph convolution.
> >
> > Finally, we thank the reviewer for introducing the seminal work [1]. We have cited it in our updated manuscript to make a comparative analysis.
> >
> > [1] Li, Qimai, Zhichao Han, and Xiao-Ming Wu. "Deeper insights into graph convolutional networks for semi-supervised learning." Proceedings of the AAAI conference on artificial intelligence. Vol. 32. No. 1. 2018.
> >
> > [2] Baranwal, Aseem, Kimon Fountoulakis, and Aukosh Jagannath. "Graph Convolution for Semi-Supervised Classification: Improved Linear Separability and Out-of-Distribution Generalization." International Conference on Machine Learning. PMLR, 2021.
> >
> > [3] Baranwal, Aseem, Kimon Fountoulakis, and Aukosh Jagannath. "Effects of Graph Convolutions in Multi-layer Networks." The Eleventh International Conference on Learning Representations. 2022.

---

### Official Review · Reviewer_DNKm · 2023-10-31

**Soundness:** 3 good
**Presentation:** 3 good
**Contribution:** 3 good
**Rating:** 6
**Confidence:** 3

**Summary:**

The paper compares GCN to MLP/CNN when learning on graph data. The paper sets out to show the benefits of using the graph structure in GCNs vs learning a MLP on the node features with no diffusion of features along the edges (which is called CNN in this paper). This is done by proposing a simple generative model for graphs and considering a two layer GCN model, providing exact formulas for gradient descent training, and proving theorems about the accuracy of the trained model. The results of this GCN model are compared to equivalent results of a MLP/CNN model. This way, GCN is shown theoretically to be superior to MLPs/CNNs.

**Strengths:**

The paper treats all aspects of learning with regard to the proposed model. Namely, they derive a formula for the training process and for the resulting accuracy. Hence, the paper provides a complete theoretical analysis, which rigorously proves the benefits of GCN over CNNs (for the proposed data model). The paper is clearly written.

**Weaknesses:**

The model of the data is very specific and limited, even for a two class model. The two classes are modeled as some template signal and its negative, without noise that directly affects the signal, e.g., additive noise. The noise is supported on indices which are disjoint from the signal. It is ok to consider simplistic models when doing mathematical analysis, but this should be clear for the reader. The fact that the model is simplistic should be clarified better in the text. The motivation for choosing such a simplistic model can be explained better. Namely, to be able to clearly decouple the signal learning aspect from the noise memorization aspect of learning.

**Questions:**

In section 3.3, why do you call this model a CNN and not a MLP? You apply a MLP on each node separately, and there is no diffusion/mixing between the features of the different nodes. It is only equivalent to a 1x1 standard CNN. I would call it a MLP that treats each node as a separate data point.

Page 8, Verification via real-world data: In your model, the two features come from one template with positive and negative sign, while in the experiment you have two features that come from two different templates. Can you explain in the paper how the experiment differs from the theoretical setting (if I am not missing something)? Can you run an experiment fully modeled as the proposed data model?

Since the experiment synthetically builds graph data that corresponds to the proposed data model, it does not show that the proposed model can describe real data. Is there some real data that can be approximated by your model? Namely, can you fit the parameters of the model to data, and check its accuracy/ability to represent real data? Or is there no real data that corresponds to your mode?

---

> ### Author Response · Authors · 2023-11-19
> **Response to Reviewer DNKm [Part I]**
>
> Thank you for your thorough review and constructive feedback. We appreciate your recognition of our paper's strengths and would like to address the concerns you raised.
>
> ___
>
> > **Weakness: The model of the data is very specific and limited, even for a two class model. The two classes are modeled as some template signal and its negative, without noise that directly affects the signal, e.g., additive noise. The noise is supported on indices which are disjoint from the signal. It is ok to consider simplistic models when doing mathematical analysis, but this should be clear for the reader. The fact that the model is simplistic should be clarified better in the text. The motivation for choosing such a simplistic model can be explained better. Namely, to be able to clearly decouple the signal learning aspect from the noise memorization aspect of learning.**
>
> Thank you for pointing out the need for greater clarity regarding our data model design. We have added a discussion in the updated manuscript:
>
> Our data model is able to clearly decouple the signal learning aspect from the noise memorization aspect of learning in the presence of complex structure brought by graph convolution. This contributes to the clear theoretical analysis and understanding of the role of graph convolution.
>
> ___
>
> > **Question 1: In section 3.3, why do you call this model a CNN and not a MLP? You apply a MLP on each node separately, and there is no diffusion/mixing between the features of the different nodes. It is only equivalent to a 1x1 standard CNN. I would call it a MLP that treats each node as a separate data point.**
>
> Our choice to label the model as a CNN, rather than an MLP, is based on the specific characteristics of the data model we used, which is a two-patch model. In our network, we implement a convolution operation using a single weight across these two patches, akin to applying a filter in traditional CNNs. This approach is consistent with the related works [1,2,3], where similar network structures applying weights across multiple data patches are identified as CNNs.
>
> ___
>
> > **Question 2: Page 8, Verification via real-world data: In your model, the two features come from one template with positive and negative sign, while in the experiment you have two features that come from two different templates. Can you explain in the paper how the experiment differs from the theoretical setting (if I am not missing something)? Can you run an experiment fully modeled as the proposed data model?**
>
> In our MNIST data experiments, we acknowledge that the signal vectors are represented in different directions. However, our theoretical results are adaptable to scenarios with multiple signal vectors, as long as these vectors maintain orthogonality. This generalization is supported by similar findings in other studies [1,4,5]
>
> We have conducted a specific experiment that closely follows our proposed theoretical model, the details of which are presented in Figure 3 of our manuscript. In this particular experiment, the input features are drawn precisely as per our theoretical model.
>
> ___
>
> > **Question 3: Since the experiment synthetically builds graph data that corresponds to the proposed data model, it does not show that the proposed model can describe real data. Is there some real data that can be approximated by your model? Namely, can you fit the parameters of the model to data, and check its accuracy/ability to represent real data? Or is there no real data that corresponds to your mode?**
>
> Thank you for your question about the representation of real data by our model. We address this through two aspects:
>
> 1. **Node Feature Models in Real-World Datasets**:
>
> Real-World Dataset Parallels: We draw parallels between our model and three commonly used citation networks (Cora, Citeseer, and Pubmed). In these datasets, the bag-of-words feature representation (often one-hot encoded) ensures orthogonality. We conceptually divide words into two categories: label-relevant and label-irrelevant. For example, words like "algorithm" or "neural network" are label-relevant to the subject of computer science, while general words like "study" or "approach" are label-irrelevant. In our signal-to-noise model, $\boldsymbol{\mu}$ represents label-relevant features, while $\boldsymbol{\xi}$ represents label-irrelevant ones.
>
> 2. **Graph Data Models and SBM**
>
> The Stochastic Block Model (SBM) is widely recognized as an effective model for studying graph structures in research. It offers insights into the homophily and heterophily characteristics of graphs, aspects that are crucial in tasks like node classification [6,7].
>
> ___
>
> **References**
>
> [1] Shen, Ruoqi, Sébastien Bubeck, and Suriya Gunasekar. "Data augmentation as feature manipulation." International conference on machine learning. PMLR, 2022
>
> [2] Cao, Yuan, et al. "Benign overfitting in two-layer convolutional neural networks." Advances in neural information processing systems 35 (2022): 25237-25250.

---

> > ### Author Response · Authors · 2023-11-19
> > **Response to Reviewer DNKm [Part II]**
> >
> > [3]  Kou, Yiwen, Zixiang Chen, Yuanzhou Chen, and Quanquan Gu. "Benign Overfitting in Two-layer ReLU Convolutional Neural Networks." International Conference on Machine Learning (2023).
> >
> > [4] Chen, Zixiang, et al. "Why Does Sharpness-Aware Minimization Generalize Better Than SGD?." arXiv preprint arXiv:2310.07269 (2023).
> >
> > [5] Zou, Difan, et al. "The benefits of mixup for feature learning." arXiv preprint arXiv:2303.08433 (2023).
> >
> >
> > [6] Baranwal, Aseem, Kimon Fountoulakis, and Aukosh Jagannath. "Graph Convolution for Semi-Supervised Classification: Improved Linear Separability and Out-of-Distribution Generalization." International Conference on Machine Learning. PMLR, 2021.
> >
> > [7] Baranwal, Aseem, Kimon Fountoulakis, and Aukosh Jagannath. "Effects of Graph Convolutions in Multi-layer Networks." The Eleventh International Conference on Learning Representations. 2022.

---

> > > ### Comment · Reviewer_DNKm · 2023-11-22
> > >
> > > I thank the authors for their response. I do not have remaining concerns with the paper, and think it should be accepted. In my opinion the strengths of the paper (full rigorous theory) outweigh the weaknesses (simplistic model).

---

> > > > ### Author Response · Authors · 2023-11-22
> > > >
> > > > We sincerely thank you for your follow-up comments and the positive assessment of our paper. We are grateful for your recognition of the strengths of our work, particularly the rigor of our theoretical approach.
> > > >
> > > > It is encouraging to know that you support the acceptance of our paper.  We look forward to the possibility of our research making a meaningful contribution to the field.

---

### Official Review · Reviewer_9JJs · 2023-11-01

**Soundness:** 2 fair
**Presentation:** 3 good
**Contribution:** 3 good
**Rating:** 6
**Confidence:** 3

**Summary:**

This paper presents a novel angle to view the training set and generalization ability of GNNs compared to CNNs. It is based on a data model that combines both the signal-to-noise for the node feature part and the stochastic block model for the graph structure. For the analysis, it introduces a signal-to-noise ratio quantity to better describe the shape of the dataset, which is closer to the utility of GNNs. Experiments on simulation datasets coincide with the theoretical analysis.

**Strengths:**

- The whole paper is well-structured and problem is well formulated, especially both the training phases and generalization ability are discussed.
- Detailed and quantitative explanation are entailed along with the arguments of this paper, also the explanation of the numbers are helpful in understanding such proofs with practical impressions.
- Proof stretches are friendly to readers.

**Weaknesses:**

- I fully agree that it is a theoretical paper that sufficient assumptions are necessary, while the node feature part is somehow far from the reality, as there are no such golden methods to divide any feature into such two orthogonal groups. Therefore, more connection of the applicability of this node feature model to the real-world setting is demanded, e.g. by showing some repression/clustering results of some real-world node features on such two orthogonal part, which is not required to be identifiable.
- Homophily is considered to be a key factor for GNNs, especially unavoidable for those analytical works based on SBM, where the interclass and intraclass connections are explicitly modeled. However, this paper does not include such a discussion, but rather focuses on the distribution of the node features.
- For Figure 3 and 4, it is better to provide variance bars by repeating the experiments of several times to show the stability of the results.

**Questions:**

- For the signal-noise model of the data model part, $x^{(1)}$ means the most ``informative'' node features. And does it imply that a binary classification setting according to $y$ is sampled from $\{-1,1\}.$ Is it possible that some parts of the conclusion of the paper can be influenced in a multi-classification setting, or just for the sake of proof?
- Should the subsection of 3.4 be part of 3.3? This is a minor issue. Another small suggestion is to standardize the use of the names MLP or CNN.
- For verification on real data, how is the graph of points 1 and 2 of MNIST constructed? By the same possibility of interclass and intraclass connections?
- It would be greatly appreciated if the authors could provide some intuition behind the dimensionality of the two parts of the node features in the data model.

---

> ### Author Response · Authors · 2023-11-19
> **Response to Reviewer 9JJs [Part I]**
>
> Thank you for your thoughtful review and the constructive comments on our submission. We are pleased that you found the paper well-structured and the problem well-formulated. We appreciate your feedback and would like to address your concerns as follows:
>
> ___
>
> > **Weakness 1: I fully agree that it is a theoretical paper that sufficient assumptions are necessary, while the node feature part is somehow far from the reality, as there are no such golden methods to divide any feature into such two orthogonal groups. Therefore, more connection of the applicability of this node feature model to the real-world setting is demanded, e.g. by showing some repression/clustering results of some real-world node features on such two orthogonal part, which is not required to be identifiable.**
>
> We appreciate your concern regarding the practical applicability of our node feature model. To address this, we draw parallels between our model and three commonly used citation networks to demonstrate the correspondence:
>
> 1. **Dataset Descriptions**
>
> - The Cora dataset consists of 2,708 scientific publications classified into one of seven classes, and 5,429 links. Each publication is described by a 0/1-valued word vector indicating the absence/presence of the corresponding word from the dictionary. The dictionary consists of 1,433 unique words
>
> - The Citeseer dataset consists of 3,312 scientific publications classified into one of six classes, and 4,732 links. Each publication is described by a 0/1-valued word vector indicating the absence/presence of the corresponding word from the dictionary. The dictionary consists of 3,703 unique words
>
> - The Pubmed Diabetes dataset consists of 19,717 scientific publications from PubMed database pertaining to diabetes classified into one of three classes. The citation network consists of 44,338 links. Each publication is described by a TF-IDF weighted word vector from a dictionary comprised of 500 unique word
>
> 2. **Feature Representation and Orthogonality**:
>
> These datasets employ a bag-of-words feature representation, typically represented by one-hot vectors, thereby ensuring orthogonality between features.
> We can conceptually divide words into two categories: label-relevant and label-irrelevant. For example, words like "algorithm" or "neural network" are label-relevant to the subject of computer science, while general words like "study" or "approach" are label-irrelevant.
> In our SNM, $\boldsymbol{\mu}$ represents label-relevant features, while $\boldsymbol{\xi}$ represents label-irrelevant ones.
>
>  3. **Quantitative Measure of Orthogonality**:
>
>
> To empirically validate the orthogonality relation between features, we provide a cossin similarity measure. The features are split into two parts of equal size (excluding the final feature for odd-sized representations) and the cossin similarity is calculated::
>
> |     Dataset    |Feature Dimension | Cossin Similarity |
> |--------------- |--------------------|--------------|
> | Cora         | 1433         | 1.57*$10^{-5}$      |
> | Citeseer   | 3703        | 3.99*$10^{-6}$        |
> | Pubmed     | 500    | 2.00*$10^{-4}$          |
>
>
> Although the features are not strictly divided into label-relevant and label-irrelevant groups, the results in the table support the orthogonality relation posited in our model.
>
> ___
>
> > **Weakness 2: Homophily is considered to be a key factor for GNNs, especially unavoidable for those analytical works based on SBM, where the interclass and intraclass connections are explicitly modeled. However, this paper does not include such a discussion, but rather focuses on the distribution of the node features.**
>
>
> We appreciate your insightful suggestion regarding the inclusion of a discussion on homophily, especially in the context of Stochastic Block Models (SBM). We have now incorporated this aspect into our paper:
>
> In the SBM framework, the inter-class probability $p$ and intra-class probability $s$ are explicitly modeled, allowing us to analyze different graph structures based on the relationship between $p$
> and $s$. When $p$ is significantly greater than $s$, the graph structure exhibits homophily. This means that the labels of neighboring nodes are likely to be similar to the label of the central node. Conversely, a heterophily graph structure is observed when $s$ is significantly greater than $p$. In this situation, nodes are more likely to connect with nodes of different labels.
>
> ___
>
> > **Weakness 3: For Figure 3 and 4, it is better to provide variance bars by repeating the experiments of several times to show the stability of the results.**
>
> We appreciate your suggestion regarding the inclusion of variance bars in Figures 3 and 4. We have repeated the experiments five times and updated these figures to include variance bars. The revised figures with the added variance information are available in the updated version of our manuscript.

---

> ### Author Response · Authors · 2023-11-19
> **Response to Reviewer 9JJs [Part II]**
>
> > **Question 1: For the signal-noise model of the data model part, means the most ``informative'' node features. And does it imply that a binary classification setting according to is sampled from −1,1. Is it possible that some parts of the conclusion of the paper can be influenced in a multi-classification setting, or just for the sake of proof?**
>
> It is feasible to extend our results to a multi-classification scenario. In this extension, we could assume that each class is represented by a distinct signal vector (e.g., $\boldsymbol{\mu}_1,  \boldsymbol{\mu}_2, \cdots, \boldsymbol{\mu}_K$ for $K$ classes), with labels represented by one-hot vectors. The key condition for this extension is ensuring that each signal vector is orthogonal to the others. Under this assumption, our theoretical results can be naturally adapted to accommodate multi-class settings.
>
> ___
>
> > **Question 2: Should the subsection of 3.4 be part of 3.3? This is a minor issue. Another small suggestion is to standardize the use of the names MLP or CNN.**
>
> Thank you for your valuable suggestions regarding the organization of our paper and the standardization of terminology. We have carefully reviewed the structure and made the following modifications:
>
> - **Reorganization**: Based on your suggestion, we have integrated Section 3.4 into Section 3.3, removing the separate title of Section 3.4.
> - **Terminology Standardization**: We have also standardized the use of terms 'MLP' and 'CNN' throughout the paper to maintain consistency and avoid any confusion.
>
> ___
>
> > **Question 3: For verification on real data, how is the graph of points 1 and 2 of MNIST constructed? By the same possibility of interclass and intraclass connections?**
>
> In verifying our theoretical results on real data, specifically for points 1 and 2 of the MNIST dataset, we employed a stochastic block model to construct the graph structure. Here's how we approached it:
>
> 1. Each image in the dataset is treated as a node in the graph.
>
> 2. When constructing connections between nodes (images), we set different probabilities based on whether the images represent the same number or different numbers:
>
>      - For images with the same number (for example, both images depicting the number '1' or both depicting '2'), we set the connection probability at $p=0.75$.
>
>     - Conversely, for images depicting different numbers, the connection probability is set at a lower rate of $s=0.05$.
>
> We have added this information in our updated manuscipt.
>
> ___
>
> > **Question 4:  It would be greatly appreciated if the authors could provide some intuition behind the dimensionality of the two parts of the node features in the data model.**
>
> Thank you for your inquiry regarding the intuition behind the dimensionality of the two parts of node features in our data model. Our choice to assume equal dimensionality for both parts was primarily for simplification and clarity in our theoretical analysis.
>
> However, we recognize that this model can be extended to more complex scenarios. For example, if we were to introduce additional noise or signal patches, the dimensions attributed to noise and signal could differ. This adaptation would allow our model to better reflect real-world data scenarios, where the balance between signal and noise might not be symmetrical.

---

> > ### Comment · Reviewer_9JJs · 2023-11-21
> > **Response**
> >
> > Dear all authors,
> >
> > Thank you all for your responses. I think most of my concerns have been addressed in some way. However, I would like to emphasize two issues that need to be further clarified:
> > - Regarding the orthogonal parts of the node features, obviously the three citation networks are consistent with such a data model, but in more general real-world settings this is probably not the case, which will lead to weaker applicability of the conclusion of this paper in such scenarios. Please ask the author to justify such cases or discuss them in the limitations section.
> > - As for the discussion of homophily, I may not be able to make it clear at the review stage. My point is, in this paper, regarding the simulated dataset preserving homophily by setting significant $p$, and real-world networks are all well-known homophilic ones, I worry that the conclusions draw by this paper is related to the homophily property behind. Please the authors also justify this part.
> >
> > Since there are no other serious problems on my side, I'm willing to keep the original score.
> >
> > Best,

---

> ### Author Response · Authors · 2023-11-23
>
> Thank you for your continued engagement with our paper and for acknowledging the efforts made in addressing your initial concerns. We appreciate your constructive feedback and the opportunity to clarify the remaining issues.
>
> 1. **Orthogonal Parts of Node Features in General Settings**
>
> We have extended our analysis to measure the cosine similarity between two equal-sized parts of node features (excluding the final feature for odd-sized representations) across a wider range of datasets. This additional analysis further supports the orthogonality relation posited in our model. The results are as follows:
>
>
> |     Dataset    |Feature Dimension | Cossin Similarity |
> |--------------- |--------------------|--------------|
> | Cora         | 1433         | 1.57*$10^{-5}$      |
> | Citeseer   | 3703        | 3.99*$10^{-6}$        |
> | Pubmed     | 500    | 2.00*$10^{-4}$          |
> | Coauthor CS|  6805 | 2.28*$10^{-6}$ |
> | Coauthor Phy.| 8451 | 1.08*$10^{-6}$|
> |Amazon Comp.|  767| 9.00*$10^{-4}$ |
> |Amazon Photo|  745| 9.00*$10^{-4}$ |
> |WikiCS|  300  |  1.00*$10^{-4}$   |
>
> To provide additional context, the datasets Wiki-CS, Amazon-Computers, Amazon-Photo, Coauthor-CS, and Coauthor-Physics [1] are characterized as follows:
>
> - Coauthor CS (Computer Science) & Coauthor Physics (Coauthor Phy.): The dataset typically includes features based on the keywords of an author's papers, and the task is often to predict each author's research field or interests based on their publication record and collaboration network.
>
> - Amazon Computers & Amazon Photo: Node features are derived from product reviews, and the classification task involves predicting product categories based on the co-purchase relationships and review data.
>
> - WikiCS Node features could be derived from the text of the articles, such as word vectors. The classification task usually involves categorizing articles into different areas or subjects within Computer Science based on their content and the article network structure.
>
> 2. **Further Discussion on Homophily**
>
> Regarding your concern about homophily, we will provide a more detailed justification of how our conclusions are influenced by the homophily property in simulated datasets. We agree that understanding the role of homophily is crucial for interpreting our results accurately, and we will make an effort to address this in our discussion, ensuring a comprehensive understanding of its impact.
>
> **Reference**
>
> [1] Oleksandr Shchur, Maximilian Mumme, Aleksandar Bojchevski, and Stephan Günnemann.
> Pitfalls of graph neural network evaluation. arXiv preprint arXiv:1811.05868, 2018.
>
> Once again, we thank you for your insightful feedback and look forward to your further thoughts and comments to help us refine our work.

---

### Official Review · Reviewer_UPB7 · 2023-11-01

**Soundness:** 3 good
**Presentation:** 3 good
**Contribution:** 3 good
**Rating:** 5
**Confidence:** 4

**Summary:**

This study aims to address the knowledge gap by investigating the role of graph convolution in feature learning theory under a specific data generative model. They conduct a comparative analysis of optimization and generalization between two-layer graph convolutional networks (GCNs) and their convolutional neural network (CNN) counterparts. They indicate that graph convolution significantly improves the range of low test errors compared to CNNs. This highlights a significant discrepancy in generalization capacity between GNNs and MLPs, a conclusion further supported by our empirical simulations on both synthetic and real-world datasets.

**Strengths:**

Several theoretical results about two-layer GCNs are obtained to compare with CNNs.

**Weaknesses:**

The practical significance of the theoretical results is unclear.

**Questions:**

1. I am confused with the significance of the theoretical results provided in this work. What's the purpose? Guide to improve GCN's training efficiency? or explain why GCNs better than CNNs?

2. The main difference between GCNs and CNNs is the introduction of graph structure information, such as adjacency matrix A and Laplacian matrix L, where the theoretical results in this work have not entirely reflected the structure information (A or L), please explain this in detail.

---

> ### Author Response · Authors · 2023-11-18
> **Response to Reviewer UPB7 [Part I]**
>
> Thank you for your insightful feedback and constructive comments on our submission. We appreciate the opportunity to clarify your concerns and questions as follows:
> ___
>
> > **Q1 & Weakness: I am confused with the significance of the theoretical results provided in this work. What's the purpose? Guide to improve GCN's training efficiency? or explain why GCNs better than CNNs?**
>
> We would like to clarify that the primary purpose of our theoretical results is to explain **why Graph Convolutional Networks (GCNs) are better than Convolutional Neural Networks (CNNs) in certain contexts**, particularly when dealing with graph-structured data. In particular, we show how GNNs leverage structural information through graph convolution to achieve better feature learning compared to CNNs. Our work provides a deeper understanding of the fundamental mechanisms that contribute to the enhanced performance of GCNs over CNNs. The significance of this work can be summarized as follows:
>
> - In contrast to earlier works that primarily focused on the (non-)linear separability of the input data points for graph neural networks [1,2], our research takes a step further by investigating the actual training dynamics of GCNs. This demonstrates that a GCN can indeed be trained to separate the data, *a non-trivial achievement due to the non-convexity of the optimization problem*. Furthermore, it has not yet been studied whether a trained GCN can generalize well on test data. We successfully provide insights into the *generalization gap between GCNs and CNNs post-training, which is a major contribution of our work*.
>
> - Technically, we introduce an innovative approach to the study of “benign overfitting” [3,4,5]. By leveraging this concept, we successfully demonstrate the differences in optimization and generalization between Graph Convolutional Networks (GCNs) and Convolutional Neural Networks (CNNs). We would like to emphasize that the results in [3,4,5] cannot be directly applied to Graph Neural Networks or the context stochastic block model, as our work is specifically focused on graph convolution. To address this, we have introduced *novel techniques to overcome the unique challenges presented by the incorporation of graph structure into neural networks*. This approach not only distinguishes our work from existing studies but also significantly contributes to the understanding of how GCNs function differently from CNNs in terms of learning and generalization.
>
> ___
>
> > **Q2: The main difference between GCNs and CNNs is the introduction of graph structure information, such as adjacency matrix A and Laplacian matrix L, where the theoretical results in this work have not entirely reflected the structure information (A or L), please explain this in detail.**
>
> In this work, we consider the graph convolution formulated by $\tilde{\mathbf{D}}^{-1} \tilde{\mathbf{A}}$ with $\tilde{\mathbf{A}} = \mathbf{A} + \mathbf{I}_n$ representing the adjacency matrix with self-loop.
>
> The effect of graph convolution on the input feature is characterized by $\tilde{\mathbf{X}} = \tilde{\mathbf{D}}^{-1} \tilde{\mathbf{A}} \mathbf{X}$.
>
> We examine the feature after graph convolution. The input feature comprises two components: the signal part and the noise part. We find that the signal patch after graph convolution maintains a similar magnitude ($\ell_2$ norm) as the original signal because of homophily. However, we find that the noise patch after graph convolution is significantly reduced. Therefore, compared to CNN, GNN focuses more on signal learning than noise memorization.
>
> Finally, the difference of feature learning between GNN and CNN caused by additional graph convolution ($\tilde{\mathbf{D}}^{-1} \tilde{\mathbf{A}}$) is characterized by the difference in the generalization ability. For CNN, we find that if $n \mathrm{SNR}^q > 1$, then the test loss tends to zero (in the benign overfitting regime). For GNN, if  $n \mathrm{SNR}^q (n(p+s))^{q-2} > 1$, then the test tends to zero.
>
> By comparing the condition for achieving near-zero test error, we find that GNNs are superior as they have a wider range, described by (n(p+s))^{q-2}, where n(p+s) is the expected degree of the graph. Through this finding, we demonstrate how graph convolution helps GNN achieve better generalization compared to CNN.

---

> > ### Author Response · Authors · 2023-11-18
> > **Response to Reviewer UPB7 [Part II]**
> >
> > **References**
> >
> > [1] Baranwal, Aseem, Kimon Fountoulakis, and Aukosh Jagannath. "Graph Convolution for Semi-Supervised Classification: Improved Linear Separability and Out-of-Distribution Generalization." International Conference on Machine Learning. PMLR, 2021.
> >
> > [2] Baranwal, Aseem, Kimon Fountoulakis, and Aukosh Jagannath. "Effects of Graph Convolutions in Multi-layer Networks." The Eleventh International Conference on Learning Representations. 2022.
> >
> > [3] Frei, Spencer, Niladri S. Chatterji, and Peter Bartlett. "Benign overfitting without linearity: Neural network classifiers trained by gradient descent for noisy linear data." Conference on Learning Theory. PMLR, 2022.
> >
> > [4] Cao, Yuan, et al. "Benign overfitting in two-layer convolutional neural networks." Advances in neural information processing systems 35 (2022): 25237-25250.
> >
> > [5]  Kou, Yiwen, Zixiang Chen, Yuanzhou Chen, and Quanquan Gu. "Benign Overfitting in Two-layer ReLU Convolutional Neural Networks." International Conference on Machine Learning (2023).

---

### Official Review · Reviewer_LXCw · 2023-11-01

**Soundness:** 4 excellent
**Presentation:** 3 good
**Contribution:** 2 fair
**Rating:** 5
**Confidence:** 5

**Summary:**

The paper provides grounded theoretical exploration to analyze the characteristics, effectiveness, and superbness of GCNs (compared with other common modules), revealing the role that graph convolution plays. It has several interesting findings, especially the ones about test erros.

**Strengths:**

(placeholder for future edit, please allow me extra hours to finish the writing.)

**Weaknesses:**

(placeholder for future edit, please allow me extra hours to finish the writing.)

**Questions:**

(placeholder for future edit, please allow me extra hours to finish the writing.)

---

### Author Response · Authors · 2023-11-21
**Global Response to All Reviewers**

Dear reviewers,

We are immensely grateful for your thoughtful and constructive feedback on our submission. It is truly encouraging to receive recognition for various aspects of our paper, including its **rigorous theoretical results** (Reviewer UPB7, Reviewer DNKm), **well-written presentation** (Reviewer 9JJs, Reviewer DNKm)，**comprehensive theoretical analysis** (Reviewer 9JJs, Reivewer DNKm, Reviewer Q462)，**reader-friendly proofs** (Reviewer 9JJs), **clear simulation experiments** (Reviewer Q462), and **import motivation** (Reviewer Q462).

In response to your valuable insights, we have made the following modifications to our manuscript (all updates are highlighted in red):

1. Added a discussion on the stochastic block model, as suggested by Reviewer 9JJs.

2. Updated Figures 3 and 4 to include error bars with five runs, following Reviewer 9JJs's recommendation.

3. Included information on edge probability in Figure 4 to address questions from Reviewer 9JJs.

4. Added a discussion in Appendix D.2 to connect our theoretical data model with real-world datasets, responding to concerns raised by Reviewers DNKm and 9JJs.

Additionally, we wish to emphasize the significance of our theoretical work to Reviewers UPB7 and Q462:

Our work aims to elucidate why Graph Convolutional Networks (GCNs) are superior to Convolutional Neural Networks (CNNs) in specific contexts, particularly with graph-structured data. This is a significant topic as earlier works have not demonstrated the benefits of graph convolution, considering the training process (optimization) and generalization to test data. Our research provides non-trivial insights into the generalization gap between GCNs and CNNs post-training, a major contribution of our work. We have introduced novel techniques to address the unique challenges posed by integrating graph convolution into feature learning theory. Given the strong connection between our data model and real-world datasets, our work offers a deeper understanding of the fundamental mechanisms contributing to GCNs' enhanced performance over CNNs.

We have made earnest efforts to address the primary concerns raised and look forward to further feedback from the reviewers to enhance the quality of our manuscript.

---

### Meta-Review · Area_Chair_AGwk · 2023-12-05

**Metareview:**

Note that I discarded one of the review proposed (score: 5) since the reviewer did not argument it.

The submission presents a theoretically sound and well-presented study comparing GCNs and CNNs. While the paper presents a rigorous theoretical analysis and clarity, there are substantial concerns regarding its practical significance. The highly specific and somewhat unrealistic data model limits the applicability of the findings. A reviewer was particularly concerned on the impact of homophily on the results (both in theory and in practice).

**Justification For Why Not Higher Score:**

Lack of practical impact

**Justification For Why Not Lower Score:**

N/A

---

### Decision · Program_Chairs · 2024-01-16

Reject